# Telomere Shortening and Psychiatric Disorders: A Systematic Review

**DOI:** 10.3390/cells10061423

**Published:** 2021-06-07

**Authors:** Pedro A. Pousa, Raquel M. Souza, Paulo Henrique M. Melo, Bernardo H. M. Correa, Tamires S. C. Mendonça, Ana Cristina Simões-e-Silva, Débora M. Miranda

**Affiliations:** 1Interdisciplinary Laboratory of Medical Investigation, Faculty of Medicine, Federal University of Minas Gerais (UFMG), Belo Horizonte, Minas Gerais 30130-100, Brazil; pedropousa17@gmail.com (P.A.P.); raquelsouza@gmail.com (R.M.S.); phmm2000@gmail.com (P.H.M.M.); berncmendes@gmail.com (B.H.M.C.); tamiressara@hotmail.com (T.S.C.M.); acssilva@hotmail.com (A.C.S.-e.-S.); 2Department of Pediatrics, Laboratory of Molecular Medicine, UFMG, Belo Horizonte, Minas Gerais 30130-100, Brazil

**Keywords:** psychological distress, telomere, traumatic stress disorder, depression, anxiety

## Abstract

Telomeres are aging biomarkers, as they shorten while cells undergo mitosis. The aim of this study was to evaluate whether psychiatric disorders marked by psychological distress lead to alterations to telomere length (TL), corroborating the hypothesis that mental disorders might have a deeper impact on our physiology and aging than it was previously thought. A systematic search of the literature using MeSH descriptors of psychological distress (“Traumatic Stress Disorder” or “Anxiety Disorder” or “depression”) and telomere length (“cellular senescence”, “oxidative stress” and “telomere”) was conducted on PubMed, Cochrane Library and ScienceDirect databases. A total of 56 studies (113,699 patients) measured the TL from individuals diagnosed with anxiety, depression and posttraumatic disorders and compared them with those from healthy subjects. Overall, TL negatively associates with distress-related mental disorders. The possible underlying molecular mechanisms that underly psychiatric diseases to telomere shortening include oxidative stress, inflammation and mitochondrial dysfunction linking. It is still unclear whether psychological distress is either a cause or a consequence of telomere shortening.

## 1. Introduction

Telomeres are repetitive, long sequences of DNA located at the termini of human chromosomes. Telomere hexanucleotides (5′-TTAGGG-3) repeats, for instance, weigh around 7–24 kb [1]. They are also rich in repetitive single-stranded guanine (G) sequences, which may range from 75–200 nucleotides at the chromosomal extremity. These single-stranded sequences are known as “overhangs”, which are vulnerable to enzymatic activity [2]. To reduce this vulnerability, 3′G inserts itself back into double-stranded DNA and forms structures known as t-loops [2,3]. The t-loops are stabilized by the six sheltering complex proteins (TRF1, TRF2, TIN2, POT1, TPP1 and RAP) [2,3,4].

The length of the telomeres varies with the type of cell. Nevertheless, the TL among different types of cells often shows a positive association, which means that a reduction in leukocyte telomere length usually means that other types of cells in the same body probably also have a related reduced value. This is the reason why white blood cells are used as biomarkers in clinical studies. Surprisingly, race must be taken into account, as people of African ancestry have been found to possess longer telomeres lengths [5].

During cell replication, the enzymes that duplicate genetic material are unable to perform their activity all the way to the end of the DNA [6]. Since chromosomes get shorter after each cell division, the main function of telomeres is to allow the cells to divide without losing their coding DNA regions [6]. When telomeres get too short, cells usually undergo the senescence stage, in which mitosis does not occur anymore. Therefore, it is well-known that telomere length is a marker of cell aging [7].

Past studies have established that telomere length (TL) is associated with some chronic diseases, including diabetes [8] and cardiovascular disorders [9]. A handful of case–control studies have already been conducted in order to assess the association between psychological distress, including depression and anxiety [10], and telomere shortening. These studies have evaluated many different outcomes and, thus far, there is no scientific consensus on the matter [11].

Our study was designed to gather as much data from previous studies as possible. Our aims are: (1) To perform a systematic review of the existing data; (2) To answer whether it is possible to establish a link between TL and disorders related to psychological distress, including depression, anxiety and PTSD; (3) To unveil the possible pathways of molecular mechanisms behind the telomere shortening in mental health disorders.

## 2. Methodology

### 2.1. Outcome

The expected outcome of this study is to review and analyze qualitatively the influence of distress-related disorders including depression, anxiety and PTSD on TL and to unveil the potential mechanisms underlying the telomere shortening.

### 2.2. Design

This systematic review was performed according to Preferred Reporting Items for Systematic Reviews and Meta-Analyses (PRISMA) [12] and has been registered in PROSPERO (International prospective register of systematic reviews) under the ID CRD42021229258.

### 2.3. Search Strategy

Initially, search queries (Figure 1) were created to link generic keywords associated with psychological distress-related disorders to telomere length. Our systematic search used the following descriptors (and MeSH analogs): “Traumatic Stress Disorder” or “Anxiety Disorder” or “depression”. For telomere length, we used the descriptors “cellular senescence”, “oxidative stress” and “telomere”. This search was performed at PubMed, Cochrane Library and ScienceDirect databases.

### 2.4. Inclusion and Exclusion Criteria

Study inclusion criteria were as follows: (i) Original papers; (ii) Written in English; (iii) Conducted on human subjects with proper control groups; (iv) Clearly evaluated TL on patients diagnosed with depression, anxiety or PTSD; (v) Published before 3 December 2020. Studies without a control group or those using animal models were excluded.

### 2.5. Study Selection

After an initial abstract/title screening, two independent authors (PAP and RMS) selected potential articles on the PubMed database and two independent authors (PHMM and TSCM) selected articles from the Cochrane Library and ScienceDirect databases. These initially screened studies were fully analyzed independently by the four authors (PAP, RMS, PHMM and TSCM) and any discrepancies related to the final decision were discussed among them. If a consensus was not achieved, then we referred to a fifth author (ACSS). Our study selection process counted with assistance from the Rayyan website and a mobile application [13].

### 2.6. Data Extraction

PAP, PHMM and RMS individually extracted data from the selected studies and an independent author (BM) reviewed all selected studies and the table to avoid any errors in data extraction.
Study identification: first author, publication date, country and type of study;Participants: age, gender and sample size;Variables: main comorbidities and lifestyle factors associated, race/ethnicity, psychiatric diseases (including severity and methods for diagnosis), medication and telomere measurement method.

### 2.7. Quality Assessment

All the included studies in this systematic review were evaluated according to the Newcastle–Ottawa Quality Assessment Scale (NOS) [14]. Studies were categorized into case–control studies, cohort studies and cross-sectional studies. Interestingly, the NOS guidelines assess only case–control studies and cohort studies—which was unfortunate, considering the high number of cross-sectional studies that we retrieved. Therefore, the authors designed an adapted protocol for this category using a NOS-adapted scale as a reference (Appendix B, Appendix C and Appendix D) [15]. All studies were scored by a “star-system”, in which the higher the number of stars, the better the quality. Each category of evaluation has a pre-established maximum quantity of possible stars: (i) Selection quality may sum up to four stars total; (ii) Comparability totalizes two stars only; (iii) Exposure and outcome may sum to three stars each. Our full quality assessments are displayed in Appendix A.

## 3. Results

We gathered an initial cohort of 540 studies from PubMed, 351 from Cochrane Library, and 94 from ScienceDirect (Scopus). Duplicates were promptly removed, which left us with a total of 808 studies. Our abstract/title screening filtered out 640 studies, totalizing then 168 studies to be fully evaluated. Our full-test analysis excluded 112 articles for meeting our inclusion criteria (Figure 2). A total of 56 articles were included in our qualitative analysis. Overall, we extracted data from a population of approximately 113,699 patients. Articles were from: (i) The USA (26 studies); (ii) The Netherlands (8 studies); (iii) Germany (5 studies); (iv) Sweden (3 studies); (v) United Kingdom (3 studies); (vi) Canada (2 studies); (vii) China (2 studies); (viii) France (2 studies); (ix) Israel (1 study); (x) Spain (1 study); (xi) India (1 study); (xii) Saudi Arabia (1 study); (xiii) Colombia (1 study); (xiv) Denmark (1 study); (xv) South Africa (1 study); (xvi) Croatia (1 study); (xvii) Armenia (1 study); (xviii) South Korea (1 study). Some articles were from more than just one country.

### 3.1. Telomere Length and Depressive Disorders

Out of the 56 eligible studies, 42 were on depressive disorders, including 98,564 patients [11,16,17,18,19,20,21,22,23,24,25,26,27,28,29,30,31,32,33,34,35,36,37,38,39,40,41,42,43,44,45,46,47,48,49,50,51,52,53,54,55,56]. Study designs included 20 case-control studies, 14 cross-sectional studies, and 8 longitudinal studies. Twenty-nine studies (91,095 patients) reported a negative association between TL and depression [16,18,19,23,24,25,26,27,28,29,31,32,33,34,38,40,41,42,43,44,45,46,47,48,49,52,53,54,55,56]. Nine studies (5188 patients) did not observe association of TL and depressive disorder [11,17,20,21,30,35,37]. Finally, four studies (2281 patients) came to inconclusive conflicting results [22,25,39,50].

TL measurement unit was mean TL (mTL) in 20 studies, relative telomere length (RTL) in other 20 studies and salivary telomere length (STL) in one study. One study did not specify the units.

Source tissues were: (i) Leukocytes (35 studies); (ii) Saliva samples (3 studies); Blood (1 study); White matter oligodendrocytes (1 study); Brain tissue (1 study); Tissue not specified (1 study). These findings are summarized in Table 1.

Conflicting results around the influence of the duration MDD on TL were also reported. In general, TL was associated with major depression disorder (MDD). There were some conflicts as to whether the duration of MDD had an important role on TL. While some studies suggested [23,24,40,41] that shorter TL was a consequence of long-term exposure to MDD, other studies reported that adolescents with MDD had shorter TL [50] and that only MDD younger adults had shorter TL [31,32,33,34,35,36].

### 3.2. TL and PTSD

Thirteen studies assessed TL on 5237 PTSD patients [22,54,57,58,59,60,61,62,63,64,65,66,67] (Table 2). Nine studies were cross-sectional articles, two studies were case–control articles and two studies were longitudinal articles. Six studies (3980 patients) established a negative association between TL and PTSD [60,62,63,64,65,67], three studies (649 patients) did not observe association of TL and PTSD [22,54,57], one study (128 patients) found a positive association between TL and PTSD [61] and three studies (480 patients) showed either conflictual results or provided association with other variables [58,59,66]. The TL measurement unit was R in eight studies and mTL in three studies. Source tissues were leukocytes in 12 studies and peripheral blood in one study.

The studies that did not find any association between TL and PTSD were conducted in military personnel [22,54,57,61]. The association between trauma severity with or without PTSD diagnosis, hostility, early trauma, global psychopathological severity and TL were also observed [22,57,58,59,66].

### 3.3. TL and Anxiety Disorders

A total of 11 studies (11,237 patients) assessed the relationship between anxiety and TL [11,21,26,31,34,35,42,68,69,70,71]. Four studies were cross-sectional articles, four studies were case–control articles and three were longitudinal studies (Table 3). Seven studies (9103 patients) observed a negative association between TL and anxiety [21,26,31,34,35,68,69], while only one study observed a positive association (132 patients) [42]. One study (1164 patients) reported conflicting results [11] while the two last studies (838 patients) did not observe any association [70,71].

The TL measurement unit was mTL in seven studies and RTL in four studies. All studies used leukocytes to measure TL.

Most studies suggest that current anxiety disorder is associated with shorter TL [21,26,31,34,35,68,69]. This relationship likely represents one’s predisposition to develop anxiety as a long-term consequence of TL [21,26].

### 3.4. Quality Assessment

A total of 19 studies were assessed by the guidelines of the NOS scale for case–control studies [14]. Regarding the selection quality assessment, only two studies scored four stars (∗∗∗∗), eight studies scored three stars (∗∗∗), six studies scored two stars (∗∗) and three studies scored only one star (∗). Regarding our comparability quality assessment, 13 studies scored two stars (∗∗) and six studies scored one star (∗). According to our outcome quality assessment, six studies scored three stars (∗∗∗), 11 studies scored two stars (∗∗) and two studies scored one star (∗). Most of these studies did not discuss the representativeness of the presented cases and did not include information regarding the recruitment of controls. All case–control studies scored at least one one-star score.

The NOS scale for cohort studies was applied to assess the quality of nine longitudinal studies [14]. In the selection quality assessment, three studies scored four stars (∗∗∗∗), five studies scored three stars (∗∗∗) and one study obtained two stars (∗∗). Our comparability quality assessment classified six studies as two stars (∗∗) and three studies as one star (∗). Next, about our outcome quality assessment, four studies scored three stars (∗∗∗), four studies scored two stars (∗∗) and one study scored one star (∗). Again, the lack of a discussion about the representativeness of the psychiatric groups was one of the main issues. Additionally, many studies failed at reporting the TL at the beginning of the studies. All longitudinal studies scored at least one one-star score across the categories.

Twenty-five cross-sectional studies were assessed for quality by our adapted version of the NOS scale for cross-sectional studies [15]. Regarding the selection quality assessment, seven studies scored four stars (∗∗∗∗), seven studies obtained three stars (∗∗∗), seven studies scored two stars (∗∗) and four studies scored only one star (∗). Our comparability quality assessment classified 20 studies as two stars (∗∗) and five studies as one star (∗). Lastly, our outcome quality assessment classified eight studies with three stars (∗∗∗), and 17 studies with two stars (∗∗). Several studies had small sample sizes and did not provide a proper description of non-respondents. The lack of confidence interval analyses was also another limitation. All cross-sectional studies scored at least one one-star score across the categories. Appendix A present our full quality assessment.

## 4. Discussion

In the present study, we reviewed whether TL is associated with distress-related psychiatric disorders (depressive and anxiety disorders and PTSD). The potential association between anxiety, depression and PTSD with telomere shortening should consider which features and mechanisms were common among these disorders to result in telomere erosion. The common aspect of these disorders is psychological distress, which includes the experience of suffering, and behavioral, emotional and thought problems, including the cultural varieties of the psychiatric diagnosis in diverse cultures. Under these assumptions, we discuss the common features and the mechanisms potentially impacting telomere length.

### 4.1. Risk Factors and Telomere Length

Many studies identified an association between distress and telomere shortening. Distress during pregnancy catalyzes telomere erosion in the fetus—and predisposes telomere erosion during childhood and adulthood [71]. Additional risk factors for telomere erosion include socioeconomic vulnerability [72], obesity [73] and smoking [74]. Interestingly, adulthood cellular senescence—and subsequent telomere erosion—was associated with early childhood trauma (<5 years of age) [75,76,77]. Early telomere shortening, due to distress, possibly leads not only to the emergence of chronic diseases but also to mental illnesses [78,79].

The cumulative effect of risks explains why telomere shortening can be connected with early life stress vulnerability through affecting telomerase activity, which can possibly influence the later emergence of chronic disease [78] and mental illness, which is, in turn, associated with accelerated cellular senescence [79]. In this context, active and healthy lifestyles, including healthy diets [80], adequate sleep [81] and meditation [82] may attenuate age-related TL shortening [83,84]. Accordingly, longer leukocyte telomeres are associated with longer lifespans [85].

Distress is directly related to inflammation [86], oxidative stress [87] and endocrine alterations [88], all of which are linked to telomere shortening. Healthier lifestyles, however, reduce oxidative stress, inflammation and telomere erosion [89].

### 4.2. Psychological Distress-Related Diseases and TL

TL has been associated with psychiatric disorders [10,90,91,92,93,94]. Darrow and colleagues observed a robust association between psychiatric diseases and shorter telomeres in their meta-analysis, including significant size effects for PTSD, depression and anxiety [90]. Three additional meta-analyses reported an association between short telomeres and depression [91,92,93]. Finally, one meta-analysis observed an association between PTSD and shortened TL while another meta-analysis reported an association between anxiety and shorter telomeres [10,94].

Puterman and Etel [95] proposed a lifespan model of stress-induced cell aging. Stressors mediate the behavioral and psychosocial traits of individuals, establishing two features: multisystem vulnerability, which leads to faster aging and shorter cellular lifespan, and multisystem resilience, which results in slower aging and longer lifespan [95].

Even though the relationship between psychiatric disorders and telomere shortening is well-established in the literature, the studies neither assessed the nature of this association nor investigated variables. In this regard, the present review aimed not only at investigating whether telomere length is associated with psychological distress, but also at identifying which variables (such as molecular mechanisms) might underpin this association.

#### 4.2.1. MDD and TL

MDD is currently described as persistent sadness and a lack of interest or pleasure in previously rewarding and enjoyable activities. It is estimated that this disorder affects over 322 million people worldwide, considered one of the leading causes of disability and increasing in prevalence [96,97]. Depression and other mental health disorders are raising awareness worldwide due to increasing prevalence rates, as the total estimated number of people living with depression increased by 18.4% between 2005 and 2015 [97].

MDD has a complex etiology associated with a multidimensional nature including emotional, behavioral, physical and cognitive aspects [98]. MDD has clinically heterogeneous patterns and associations with other disorders, causing methodological limitations to the consistency, coherence, validity and utility of research findings [99]. Therefore, several articles found in our systematic search showed conflicting results and the main implications and limitations will be later discussed.

While clinical information provides paradoxical and heterogeneous aspects of MDD, cellular and molecular mechanisms of the disease are advancing towards a unified model for understanding depression [100]. In patients with MDD, the corticolimbic system is deregulated and molecular pathways related to inflammatory mediators, oxidative stress, DNA damage, DNA repair and mitochondrial dysfunction might interact in the pathophysiology of the disorder [10,100]. In this sense, the study of Dean and Keshavan [101] suggests that depression has several endophenotypes, associated with distinct pathophysiological mechanisms, which can explain complex and variable clinical features. However, the authors state that these mechanisms have reciprocal interactions [101]. Therefore, molecular mechanisms of depression represent an important aspect of the disease, acting not only as causative agents of the disease but also as biomarkers of depression.

Patients with MDD patients are at greater risk for developing aging-related somatic conditions [25,102]. As mentioned, TL is a biomarker of aging, being associated with various aging-related somatic diseases [7,103]. Hence, it is reasonable to hypothesize that the shortening of TL might be associated with MDD. Our systematic search found several articles that assessed the association between TL and MDD symptoms. Despite most articles and even a meta-analysis supported the association between shorter telomeres and MDD, the heterogeneity of the methodology among the studies precludes a conclusion [90,91,92,93]. There are very few studies on the nature of this relationship. Findings from the Heart and Soul Study [39] suggest that multisystem resilience, stronger social connections, greater physical activity and better sleep quality modulate the association between telomere shortening and depressive disorders. Hence, MDD-related biological outcomes should be analyzed under a psychosocial–behavioral context, as this context shapes the nature of the direct relationship. A large cohort study conducted over 10 years showed that severe depressive symptomatology is associated with TL shortening, but shortening rates do not follow disease progression [25]. In this sense, Verhoeven and colleagues proposed two hypotheses: (i) telomere shortening is a consequence of depressive disorders, acting as a long-term cellular scar, (ii) and shorter TL is a risk factor for depression development, and longer TL attrition rates do not increase after disease establishment [25,26].

In summary, the findings of our systematic review suggest that current evidence supports cross-sectional association of shorter TL and depressive disorders [11,16,18,19,23,24,26,27,28,29,31,32,33,34,38,39,40,41,42,43,44,45,46,47,49,51,52,54,56]. However, some studies propose that complex variables related to psychological–behavioral aspects are more likely to drive this relationship than the disease itself [22,39,42,104]. When it comes to assessing the nature of this relationship, telomere shortening should not be considered under a within-person perspective, but rather under a between-person one, as TL does not follow symptomatology progression [25,26]. In addition, MDD is a complex heterogeneous disease and further studies are necessary to elucidate its relationship with TL.

#### 4.2.2. PTSD and TL

PTSD is a condition that results from traumatic events. The symptoms of PTSD include intrusive memories, hypervigilance, mood disorders and emotional withdrawal [105]. The majority of individuals who experienced traumatic events do not develop PTSD. The number of exposures and the severity of the traumatic events are associated with both the development and the severity of PTSD [105]. PTSD individuals display central noradrenergic hyperactivation and abnormalities in the hypothalamic–pituitary–adrenal (HPA) axis—thus resulting in a dysregulated stress response via unbalanced cortisol release [105,106].

The disease burden of PTSD is very high. PTSD has been linked to earlier mortality and aging-related comorbidities, such as cardiovascular disease, type 2 diabetes mellitus and dementia [107]. The accelerated aging and the mechanisms related to telomere shortening in PTSD, such as inflammation and oxidative stress, made plausible the relationship between PTSD and TL.

We analyzed 13 articles to assess the relationship between PTSD and telomere length. The findings were somewhat confusing and the studies methodologically heterogeneous, with variability in the technique of TL measurement and PTSD diagnosis. Most studies were cross-sectional and conducted on the military population, in which the overall prevalence of PTSD ranged from 6 to 13% [105,108,109,110].

The studies presented mixed findings in regard to the relationship between TL and PTSD [22,54,57,61]. Studies with military personnel did not find any association between TL and PTSD [22,54,57,61]. However, many of these studies had small sample sizes and lacked uniformity in regard to experimental design [22,54,61,64,65]. Studies that were conducted on non-military populations reported significant associations between TL shortening and PTSD [60,62,63].

In summary, although mixed findings were found, there seems to be an association between PTSD and the shortening of TL. However, the nature of this relationship remains unclear and future research is needed to determine whether mechanisms associated with the pathophysiology of PTSD promote telomere shortening and if the shortening of TL is a risk factor for PTSD or if both PTSD and telomere shortening are consequences associated with trauma [60,63,67].

#### 4.2.3. Anxiety and TL

Anxiety is defined as a feeling of inner anguish [111]. Anxiety may lead to obsessive behaviors such as nail-biting, leg bouncing (restless legs syndrome) or even stress-induced gastritis [112,113]. What differs anxiety disorders apart from benign anxiety is that anxiety disorders persist excessively and may cause a wide range of symptoms that may reduce the quality of life [111].

Anxiety disorders have been associated with dysfunctional processing of fear and dread by the amygdala, two aggregates of neurons located within the temporal lobe [111]. The limbic system, of which the amygdala is part, triggers autonomic responses through the hypothalamus, which results in cortisol release [114,115,116]. Although we observed many conflicting findings, most studies demonstrate a negative association between TL and anxiety [21,26,31,34,35]. One interesting study suggested that shorter TL could be a biomarker of later onset of anxiety [26].

Two studies did not report any association between anxiety disorders and TL [35,70]. However, one of these studies only included one anxiety disorder: panic syndrome [70]. The studies did not take into account other types of anxiety disorder including Generalized Anxiety Disorder, Agoraphobia or Social Anxiety. Likewise, the remaining study only included elderly participants [35]. Due to this reason, their findings are restricted to just one age group.

In addition, the nature of the association between TL and anxiety requires further investigation, as only one study assessed this issue [26]. Verhoeven et al. [26] suggest that it is not anxiety that causes faster telomere shortening, but instead, shorter TL could be a risk factor for the future onset of anxiety. In a longitudinal evaluation after 6 years, the authors did not find that anxiety accelerated the natural process of telomere shortening. On the other hand, individuals who already possessed shorter TLs were those most likely to suffer from an anxiety disorder.

### 4.3. Mechanisms Underlying TL Shortening and Psychological Distress-Related Diseases

Stress is defined as a homeostatic imbalance, which results in behavioral, chemical and psychiatric alterations. In an effort to restore physiological conditions, the human body responds with behavioral, chemical and psychiatric reactions [117]. Under stress, neuroendocrine, cardiovascular and psychological mechanisms are triggered, causing adrenergic hormones release, tachycardia, anxiety and mood swings [114,118,119,120,121,122]. Psychiatric disorders typically affect cognition, humor and behavior, and share common distresses [122].

Biological agents and environmental conditions that cause stress are called stressors. Stressors have an impact on the release and uptake of neurotransmitters, activating or inhibiting certain brain areas. The HPA axis plays a central role in the stress response, including cortisol release. After the arrival of stressful stimuli, the axis is activated by the release of adrenocorticotropic hormone (ACTH) secretagogues, corticotropin-releasing factor (CRF) and arginine-vasopressin (AVP) by neurosecretory neurons in the parvocellular component of the paraventricular nucleus of the hypothalamus (PVN) [123]. The long-term HPA activation and chronic cortisol release cause neurostructural changes in several brain regions, including the hippocampus, the prefrontal cortex (PFC) and the amygdala—which may even contribute to stress maladaptation faced by several individuals. This process may have a role in the perpetuation of the hypercortisolism state that characterizes chronic stress. It also contributes to the maladaptation by some individuals to stress [124,125]. However, these responses also vary according to the duration of the stress exposure, the long- and short-term consequences of the phenomenon and the related neuronal circuitries [115,116,125].

Facing stress not only impacts the overall maintenance of the HPA axis but also weakens our immune system by downregulating glucocorticoid receptors in leukocytes [126,127,128,129]. It has been demonstrated that the overactivity of the sympathetic nervous system may chronically affect the immune system. The reason for this effect remains obscure, although it has been suggested that the downregulation of glucocorticoid receptors in leukocytes may be responsible [129]. Some studies have shown that chronic stress also inhibits neurogenesis and is related to a higher incidence of neuropsychological disorders, such as depression, anxiety and Alzheimer’s disease [130,131,132].

Due to stress, depression and anxiety, the release of stress hormones, such as glucocorticoids, can promote hippocampal lesions by disrupting cellular metabolism and altering synaptic structures in specific areas. These mechanisms enhance levels of cell damage in chronic stress–depression–anxiety syndromes, elevating oxidative stress load and promoting hippocampal lesions [124].

The HPA axis relation to psychological distress-related diseases seems to differ from one condition to another. While depression is associated with hyperactivity of the HPA axis, anxiety disorders present subtle and less consistent alterations, displaying different patterns of HPA activity. Differences in HPA axis activation were also found in PTSD [123].

#### 4.3.1. Oxidative Stress

Oxidative stress is characterized by an imbalance between reactive oxygen/nitrogen species (ROS/RNS) and antioxidants [133]. ROS are essential second messengers in many intracellular signaling cascades [134]. When endogenous systems fail to neutralize these species, key biomolecules become a target of oxidative attack—even resulting in cell death [133,134]. Oxidative imbalance is involved in the pathophysiology of over 100 diseases [133,135,136].

Glucocorticoid release, such as cortisol, affects the expression of antioxidant genes that play a role in the glutathione (GSH) redox-cycle: a mechanism to detoxify cells and prevent the formation of free-radicals, which may damage telomeres [137,138]. Psychological distress-induced glucocorticoid release increases oxidative stress, which contributes to telomere shortening. Additionally, the brain is known to be the most vulnerable organ to oxidative damage, which may partially explain the association of degenerative and psychiatric diseases to oxidative stress [138,139]. For example, pyramidal cells from the hippocampal dentate gyrus—the brain region associated with the etiology of depression and memory decline—are damaged by oxidative stress [138].

Comparative studies [136] in vertebrates suggest that glucocorticoids may influence the expression of antioxidant genes involved in the glutathione (GSH) redox cycle and in regulating the antioxidant enzyme GPX (glutathione peroxidase). The glutathione redox cycle is used to detoxify cells from the accumulation of hydrogen peroxide and organic hydroperoxides, whose cleavage induced by metal ions can generate very reactive free radicals that damage telomeres [136]. Moreover, glucocorticoids may also produce non-genomic effects on glutathione metabolism by increasing the metabolism and production of ROS [137]. Therefore, the release of glucocorticoids during psychological distress may increase oxidative stress and this could be one of the factors responsible for telomere shortening in psychiatric diseases.

The brain is the most vulnerable organ to oxidative damage, due to its high oxygen consumption and lipid-rich content [138]. A total of 83 degenerative syndromes and psychiatric disorders were already associated with oxidative stress, including MDD, anxiety and PTSD [139]. The hippocampus, amygdala and prefrontal cortex are related to behavioral and cognitive deficits [138]. These brain regions are also targets to oxidative stress [138]. Pyramidal cells of dentate gyrus-cornu ammonis CA3 and granule cells of the dentate gyrus (DG) are prone to suffer from damage by oxidative stress, leading to depression, and dysfunction in learning and memory [138]. The amygdala can be hyperactivated and the PFC can suffer dendritic shrinking, which can disrupt the hippocampus-amygdala projections [138]. At last, extracellular sites of glutamatergic *N*-methyl-d-aspartate receptors can be oxidized by free radicals leading to the attenuation of long-term potentiation (LTP) and synaptic neurotransmission [138]. In this regard, Ahmed and Lingner [140] suggest that telomeres are especially susceptible to oxidative damage [140]. ROS-induced damage may not be accurately repaired as double-stranded telomere DNA sequences are more susceptible to cleavage by ROS [140].

Telomere 8-oxo-guanine (8-oxoG) is considered one of the major endogenous mutagens [141]. Acute 8-oxoG formation increases telomere fragility and is associated with impaired cell growth [142]. TRF1 and TRF2 proteins protect telomeres and are involved in telomere replication and T-loop formation, respectively—and may prevent DNA damage [6]. 8-oxoG reduces TRF1 and TRF2 concentrations, resulting in telomere shortening and cell senescence [143].

Several studies report an association between oxidative stress and distress-related psychiatric diseases, such as depression, anxiety and PTSD [144,145,146,147,148,149,150,151,152,153,154,155,156]. One meta-analysis demonstrated that oxidative stress, as measured by 8-OHdG and F2-isoprostanes, is increased in depression [144]. Depressive patients have increased 8-oxoG concentrations and decreased expression of OGG1. Such alterations are reversed upon resolution of the depressive state [144,145]. PTSD patients have different concentrations of blood antioxidant enzymes as compared to healthy controls [146].

Vaváková et al. reviewed many markers of oxidative stress in MDD patients related to neuroprogression, signaling, DNA damage, prooxidant enzymes, antioxidant enzymes, antioxidants, micronutrients, inflammation, immune reaction and lipid peroxidation [145]. Patients with mood disorders had increased 8-oxo-dG and decreased gene expression levels of OGG1 during depressive episodes and these changes might be reversed by the resolution of depressive symptoms [147]. Miller and colleagues reviewed cross-sectional studies that found significant differences in blood antioxidant enzyme concentrations and OXS-related gene expression between PTSD patients and controls [147].

As both telomeres and the brain are susceptible to oxidative damage, it seems reasonable to postulate that telomeres would be shortened in nervous tissues after chronic psychological distress. However, the role of oxidative stress in distress-related psychiatric disorders cannot be inferred from alterations of peripheral parameters [148]. Only two studies of our systematic search measured TL of MDD patients in brain tissues [45,56], while the majority of the studies measured TL in leukocytes. Post-mortem analysis in the brains of MDD donors revealed shorter TL in white matter oligodendrocytes [149]. Additionally, Szebeni and colleagues’ [45] post-mortem analyses showed shorter relative telomere lengths in white matter oligodendrocytes, but not astrocytes, in MDD donors as compared to matched control donors. Gene expression levels of oxidative defense enzymes superoxide dismutases (SOD1 and SOD2), catalase (CAT) and glutathione peroxidase (GPX1) were significantly lower in the post-mortem brain tissue of MDD donors as compared to control donors [149]. Mamdani et al. observed a significant decrease in TL and altered mRNA levels of genes involved in neuroprotection during stress response (FKBP5, CRH) in the hippocampus of MDD subjects [56]. Distress-induced hypercortisolism possibly promotes oxidative stress, which results in telomere shortening (Figure 3). Whether telomere shortening leads to psychological diseases or vice versa still remains unclear [150,151]. This finding suggests the presence of hippocampal stress-mediated accelerated cellular aging in depression.

Enhanced synaptic connectivity in the basolateral amygdala (BLA) is associated with distress-related psychiatric disorders [152]. TL is positively associated with increased activations of the amygdala and cuneus as well as increased connectivity from the posterior facial region to the ventral prefrontal cortex [153]. The mechanisms that make up the association between TL and brain functional activity remain unknown [154].

Several studies also associated oxidative stress with anxiety disorders by evaluating the expression of antioxidant genes, activity of antioxidant proteins, lipid peroxidation markers and direct and indirect intracellular ROS levels [153]. One of the pathophysiological hypotheses is that a hypercortisolism presentation in depression, anxiety and PTSD would produce oxidative stress, which, in turn, might promote telomere shortening (Figure 3). However, it is not clear if oxidative stress causes telomere erosion in CNS cells, nor is it known if ROS-induced telomere shortening in neurons and glia are a causal or contributing factor in neurological diseases [151]. Accumulating evidence suggests that by enhancing synaptic connectivity in the basolateral amygdala (BLA), symptoms of chronic anxiety and disorders such as MDD and PTSD are facilitated by enhanced availability of postsynaptic dendritic surface and synaptic inputs on principal neurons of the BLA during structural encoding of aversive experiences [152]. One study revealed that TL was positively associated with increased activation of the amygdala and cuneus, as well as increased connectivity from posterior regions of the face network to the ventral prefrontal cortex. This finding suggests that TL and genetic loading for shorter telomeres influence the function of brain regions known to be involved in emotional processing. However, precise neural mechanisms contributing to the association between TL and functional activity are still unknown [153].

#### 4.3.2. Inflammation

Inflammation is a protective physiological response involving immune cells and molecular mediators triggered by external or internal factors, such as microbial infection and ischemia, respectively [154,155,156]. Peripheral inflammation results in the production of several cytokines [155]. For example, activated macrophages stimulate produce pro-inflammatory cytokines, including TNF-α and interleukin-6 (IL-6) [157]. The association between psychiatric disorders, inflammation and telomere length is not clearly understood, but the production of pro-inflammatory cytokines seems to be the intersection between these three components.

Antidepressants display anti-inflammatory effects. Depression displays strong relation to an increase in peripheral inflammatory mediators, such as IL-6, C-reactive protein and TNF-α [157,158,159,160,161]. Studies on rodents, however, demonstrated that antidepressant treatment also reduced tissue concentrations of inflammatory molecules [162,163,164].

Clinical studies have identified the anti-inflammatory effects of antidepressant medications. Pro-inflammatory markers were found at elevated levels among depressed patients, including IL-6 [158,159,160], the C-reactive protein (CRP) [157,159,161] and TNF-α [163,165]. In rodent models of depression, the antidepressant treatment reduced tissue concentrations of inflammatory molecules [162,163,164].

Pro-inflammatory cytokines are associated with telomere shortening. One possible explanation is that inflammation increases the normal human aging process and the consequent telomere shortening [165]. Cells undergoing senesce are linked to the overexpression of inflammatory cytokines, including TNF-α and IL-6 in circulating macrophages [166]. This may explain why LTL has also been associated with TNF-α, IL-6, IL-1β and CRPs [167]. Therefore, more studies are required in order to determine if depression causes higher levels of pro-inflammatory cytokines, leading to telomere shortening, or whether the shortening of the telomere provokes higher levels of these inflammatory molecules that are involved in the pathophysiology of depression.

Moreover, inflammaging is a mechanism of aging-related alteration in intercellular communication, leading to a pro-inflammatory phenotype that can accompany aging in humans [168]. This process can be provoked by proinflammatory tissue damage, affecting the immune system and host cells. These changes lead to the activation of the NLRP3 inflammasome, ROS production and cathepsin B release from unstable lysosomes and aggregated proteins. These processes are followed by an increase in IL-1β and TNF production [168], which might participate in telomere alterations due to their direct involvement in aging. This could explain why inflammaging is essentially connected to somatic cellular senescence-associated phenotype (SASP), which has been linked not only to age-related diseases in peripheral tissues but also to neurodegenerative diseases, such as Alzheimer’s [169].

Although psychological stress increases plasma levels of pro-inflammatory cytokines, the mechanism by which psychological stress is detected by the innate immune system is not known [170]. It is hypothesized that the inflammasome has a pivotal role in linking psychological and physical stressors to the development of depression. It is suggested that these stressors would act as danger signals that are detected via pattern-recognition receptors (PRRs), such as NLRP3 and TLRs. The PRRs prepare defensive responses, inducing the synthesis of pro-IL-1β and the secretion of IL-1β [170]. In addition, NLRP3 is activated by several factors, such as the release of DNA damage into the cytosol [99], mitochondrial dysfunction and ROS [2]. Therefore, this could explain the interplay between inflammation and oxidative stress and how they would be involved in the relationship between telomere length and psychological distress.

Another intersection between psychiatric disorders, inflammation and telomere length is the involvement of cyclo-oxygenase 2 (COX 2). COX-2 is responsible for the formation of prostaglandins (lipophilic molecules produced for prolonging acute inflammatory response) [171,172]. COX-2 action is induced by proinflammatory cytokines, such as IL-1β and IL-6, that have been identified at increased levels in depressed patients [173]. Interestingly, COX 2 inhibitors have shown efficiency in enhancing the response of antidepressants. COX 2 is also continuously upregulated during aging and telomere shortening [174,175].

Inflammation is also able to regulate telomerase activity. The nuclear factor kappa B (NF-κB) regulates telomerase activity via the shelterin protein RAP1, which is involved in telomere homeostasis. Additionally, ROS displays a positive association with NF-κB during chronic inflammation [99]. This relationship seems to be tissue-specific, as telomerase activity is upregulated in immune cells and downregulated in non-immune cells. Inflammatory mechanisms are associated with leukocyte proliferation, which enhances DNA loss through successive mitosis and promotes TL shortening [176,177] (Figure 3). This idea could partially explain some of the heterogeneous results found in this systematic review since the majority of studies assessed TL from blood leukocytes.

In conclusion, the association between psychiatric disorders, inflammation and telomere length still remains to be elucidated. High stress levels might result in antigenic stimulation that imposes a demand for increased immune cell activity. Reciprocally, inflammatory mechanisms are associated with leukocyte proliferation, and consequently, enhancing DNA loss through successive mitosis [176]. Additionally, chronic inflammation may lead to oxidative stress and consequently promote telomere shortening [177] (Figure 3).

#### 4.3.3. Mitochondrial Dysfunction

Mitochondria are the main site of ROS synthesis in the cell. Unbalanced redox signaling may cause DNA damage and telomere shortening [178,179]. Mitochondrial treatment with a mitochondria-specific antioxidant, MitoQ, not only reduced telomere shortening but also increased cell lifespan under oxidative stress [180]. On the other hand, treatment with carbonyl cyanide-4-(trifluoromethoxy) phenylhydrazone (FCCP), a compound that causes mitochondrial depolarization, increased ROS synthesis and TL shortening [179,181]. Redox signaling is important for homeostasis, but its overproduction can be pathogenic, causing damage to the cell, organelles proteins and DNA. [178] This imbalance in the production of reactive species could cause oxidative and redox stress and, consequently, telomere shortening [179].

Mitochondrial dysfunction could be one of the mechanisms related to telomere shortening in psychiatric disorders. There is evidence suggesting that brain metabolism, mitochondrial functions and redox balance are impaired in psychiatric disorders [181]. Kim et al. propose that the combination of genetic and environmental factors could lead to mitochondrial damage/dysfunction and oxidative stress, resulting in psychiatric disorders [181]. A direct link between telomere dysfunction and decrease of mitochondrial mass and energy production was also proposed. Reciprocally, there is a decrease of mitochondrial DNA (mtDNA) copies in psychiatric diseases. In fact, evidence suggests a decrease in mtDNA copy number with illness progression and aging for schizophrenia, bipolar disorder, MDD and possibly autism [181].

Besides that, varied polymorphisms in the mtDNA can cause different mitochondrial oxidative phosphorylation activities. Many variants have been described as risk factors for the development of various mental illnesses. Despite different studies indicating mtDNA mutations in psychiatric disorders over the last 25 years, it is still not clear if the mutations participate in the pathophysiology of these conditions [181].

The relationship between oxidative stress and mitochondrial dysfunction results in a vicious cycle. When telomeres have reached a critical point or when their stability is threatened, classic DNA damage responses (DDR) are activated and p53 is released. A study suggests that p53 suppresses AMPK, which is responsible for activating the peroxisome proliferator-activated receptor γ (PPARγ) and co-activator (PGC-1α), the master regulator of mitochondria biogenesis [178,182]. Sahin et al. suggest that p53 binds to PGC-1α/PGC-1β promoters and represses them [183]. In both scenarios, p53′s release causes the suppression of PGC-1α and PGC-1β and their downstream targets, resulting in disturbed oxidative phosphorylation, increased ROS and diminished ATP production and gluconeogenesis [178,183]. The increased ROS can further damage telomeres and can activate the DDR, forming the vicious cycle (Figure 3) [178]. Finally, the suppression of PGC1α/β, which is the master regulator of mitochondrial biogenesis, decreases the renewal of mitochondria and contributes to the accumulation of damaged mitochondria—further damaging telomeres [178,184].

On the other hand, other studies reported that p53 increases mitochondrial function and oxidative defense mechanisms. The causes for these differences still need to be further elucidated, but the switch can be associated with ROS levels and tissue needs [178]. An association between mitochondrial dysfunction and psychiatric disorders is likely. Further research is needed to better establish this relationship and its implications in the pathogenesis of telomere shortening in individuals with mental illnesses.

## 5. Conclusions

Throughout life, we are under exposure to stressors. An unbalanced response to distress may lead to faster cellular aging. A growing body of evidence suggests that psychological distress, including depression, anxiety and PTSD, may be related to telomere shortening. Our study systematically reviewed 56 articles on the topic. Although studies were heterogeneous, most of them support the association between PTSD, depression and anxiety and shorter TL. These results were in agreement with previous reports. However, the lack of longitudinal analyses remains a big limitation for the establishment of causality.

Oxidative stress, inflammation and mitochondrial dysfunction in psychological distress-related diseases and telomere shortening are likely to mediate this relationship. Finally, these processes remain hypothetical and future investigations are still necessary to understand the effect of psychological distress on TL.

## Figures and Tables

**Figure 1 cells-10-01423-f001:**
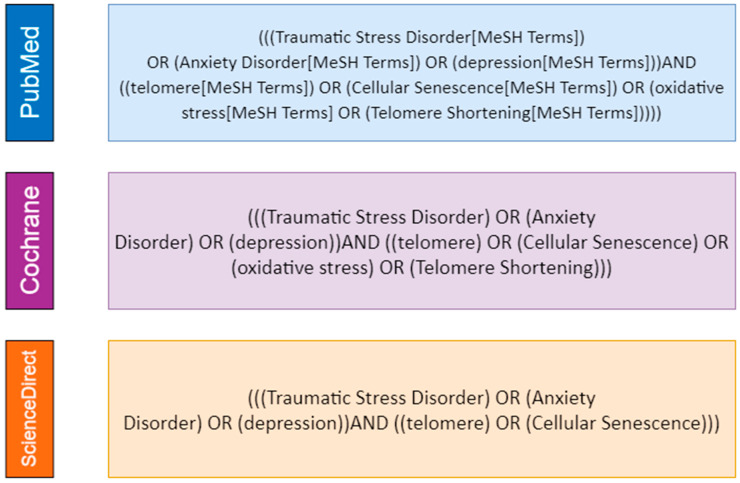
Descriptors employed in our systematic search on PubMed, Cochrane Library and ScienceDirect (Scopus) databases.

**Figure 2 cells-10-01423-f002:**
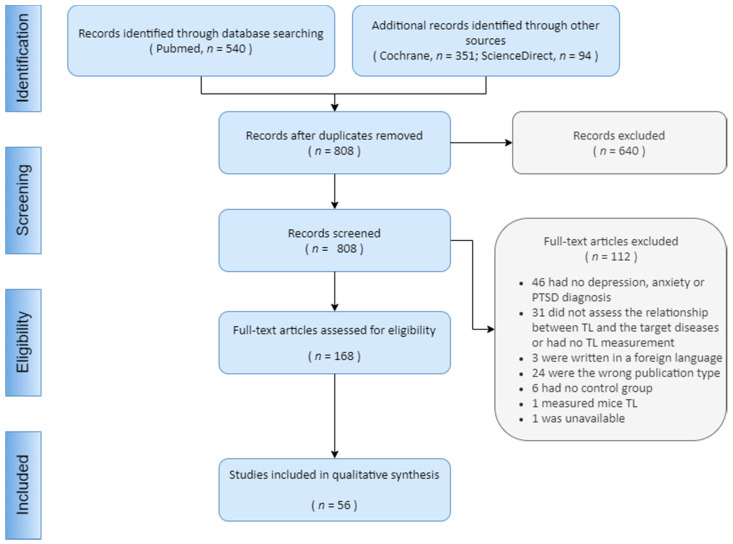
Flow diagram of study selection.

**Figure 3 cells-10-01423-f003:**
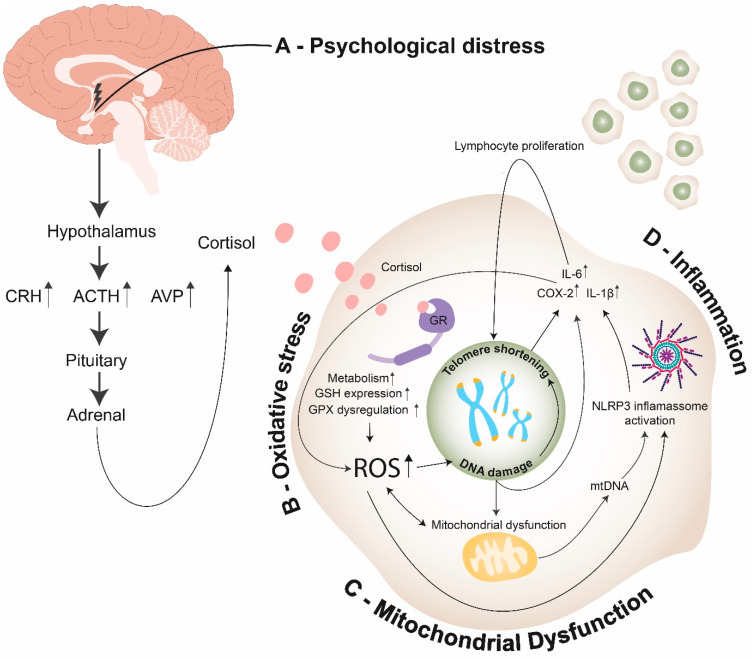
Endocrine, cellular, and molecular mechanisms linking psychological distress to telomere shortening: (**A**)—Psychological distress promotes cortisol release through the hypothalamus–pituitary–adrenal axis (HPA axis); (**B**)—In the intracellular environment, cortisol promotes the activation of GR, leading to an increase of cell metabolism, GSH expression and GPX dysregulation, thus enhancing the production of ROS. ROS can promote telomere shortening through different pathways: (1) Promoting direct DNA damage; (2) Promoting mitochondrial dysfunction, which induces an increase of ROS; (3) Promoting NLRP3 inflammasome activation. (**C**)—Mitochondrial dysfunction promotes ROS production and mtDNA release on cytosol, which results in telomere shortening through activation of NLPR3 inflammasome; (**D**)—DNA damage and NLRP3 inflammasome activation promote COX-2 and pro-inflammatory cytokine expression, resulting in the synthesis of ROS and lymphocyte proliferation—thus inducing telomere shortening. (ACTH—adrenocorticotropic hormone; CRH—corticotropin-releasing hormone; AVP—arginine vasopressin; ROS—reactive oxygen species; GR—glucocorticoid receptor; GSH—glutathione; GPX—glutathione peroxidase; mtDNA—mitochondrial DNA; NLRP3 inflammasome—NOD-, LRR- and pyrin domain-containing protein 3 inflammasomes; IL-6—Interleukin 6; IL-1β—Interleukin 1β; COX-2—cyclooxygenase-2).

**Table 1 cells-10-01423-t001:** Study characteristics, clinical and epidemiological data analysis of depressive disorders impacts on telomere lenght.

Authors	Date of Publication	Country	Study Design		Patients	Age, Years, Mean (SD)	Sex, n (%) Male	Main Comorbidities/ Lifestyle factors Associated	Race/Ethnicity	Other Associated Psychiatric Diseases in This Study	Medication	Telomere Measurement and Tissue	Measurement of Psychiatric Disorder	Level of Depression	Telomere Lenght
Garcia-Rizo et al. [16]	Feburary 2013	Spain	Case-control study	Control	70 (For telomere content, *n* = 48)	27.8 (6.8)	(62.2%)	BMI: 23.7 (2.9), Mean number of cigarettes/day: 6.2 (8.3)	NA	-	NA	mTL, Southern blot, Leukocytes	SCID	MDD (*n* = 15)	Shortened telomere are present early in the course of depression independently of the confounding. Abnormal glucose tolerance and lymphopenia were also related to MDD
				Case	15 (For telomere content, *n* = 9)	30.7 (10.0)	60%	BMI: 23.4 (4.1), Mean number of cigarettes/day: 9.9 (12.7)	NA						
Chen et al. [17]	December 2014	USA	Case-control study	Control	20	Patients were individually matched on age (±3 years), gender and ethnicity Full patients data were not available				-	NA	mTL, PCR, Leukocytes	QIDS, SCID	MDD (*n* = 20)	In healthy controls, greater ACE exposure was associated with shorter LTL but was unassociated with telomerase activity. In MDD, greater ACE exposure was unrelated to LTL but was associated with increased telomerase activity and with a higher telomerase:
				Case	20										
Tyrka et al. [18]	January 2016	USA	Case-control study	Control	113	28.5 (9.2)	50 (44.2%)	Smokers (8.3%)	White (82.3%)	Adversities, depression, PTSD and anxiety	NA	mTL, qPCR, Leukocytes	SCID, STAI, PSS, CD-RISC		Significantly higher mtDNA copy numbers and shorter telomeres were seen in individuals with major depression, depressive disorders, and anxiety disorders, as well as those with parental loss and childhood maltreatment.
				Case 1—Adverity with no psychiatric disorder	66	31.3 (11.1)	26 (39.4%)	Smokers (7.8%)	White (80.3%)						
				Case 2—Psichyatric disorder with no adversity	39	30.7 (10.4)	15 (38.5%)	Smokers (7.7%)	White (92.3%)					MDD (*n* = 6), depressive (*n* = 7)	
				Case 3—Adversity and psychiatric disorder	72	34.8 (12.0)	22 (30.6%)	Smokers (17.1%)	White (81.9%)					MDD (*n* = 7), depressive (*n* = 18)	
Prabu et al. [19]	July 2020	India	Case-control study	Control—NGT and no depression	40	48 (10)	21 (52.5%)	BMI: 26 (4.2)	NA	-	66 T2DM patients were on anti-diabetic medication alone. 14 T2DM patients were on anti-diabetic plus antihypertensive/statin medication.	RTL, rQ-PCR, blood sample	PHQ-9, PHQ-12	NA	Patients with type 2 diabetes and depression exhibited increased circulatory levels of miR-128 and serum cortisol and shortened telomeres.
				Case 1—NGT with depression	40	50 (11)	20 (50%)	BMI: 26.8 (5.8)							
				Case 2—T2DM and no depression	40	54 (6)	21 (52.5%)	BMI: 25.4 (4.9)							
				Case 3—T2DM with depression	40	54 (7)	21 (52.5%)	BMI: 25.5 (3.7)							
Vincent et al. [20]	February 2017	United Kingdom	Case-control study	Control	100	50.84 (16.89)	F:51 (51%) M:49 (49%)	BMI: 26.89 (5.39)	White	-	NA	RTL, qPCR, Leukocytes	CIS-R, SCAN	-	Shortened RTL was specifically associated with childhood physical neglect, but not the other subtypes of maltreatment or depression case/control status.
				Case	80	48.63 (13.9)	F: 52 (65%) M: 28 (35%)	BMI: 28.47 (6.87)						Mild depression (*n* = 15), moderate or severe depression (55), mixed depression/anxiety (*n* = 10)	
Hoen et al. [21]	August 2012	Netherlands	Case-control study	Control	980	53.7 (11.3)	F: 551 M: 465	Smoking (*n* = 225; 77%), Alcohol consumption (*n* = 788; 80%), Sedentarism (*n* = 505; 52%)	NA	Anxiety		mTL, PCR, Leukocytes	CIDI	NA	No association was found between depressive disorders and shorter telomeres at follow-up. Anxiety disorders predicted shorter telomere length at follow-up in a general population cohort.
				Case	97	51.3 (10.7)	F: 62 M: 36	Smokers (*n* = 32; 65%); Alcohol consumption (*n* = 78; 80%), Sedentarism (*n* = 50; 52%)			Antidepressant use (*n* = 14; 15%)				
Bersani et al. [22]	October 2015	USA	Cross-sectional study	Control	76	34.64 (9.17)	All male	Years of education (mean ± SD): 14.79 ± 2.44, current smokers (*n*): 11	Hispanic (*n* = 35); Non Hispanic (*n* = 42)*	PTSD	Statins (*n* = 2), NSAIDs (*n* = 5), antidepressants (*n* = 13), antibiotics (*n* = 1), hormone drugs for prostate cancer (*n* = 1), analgesics (*n* = 1)	RTL, PCR, Granulocytes	CAPS, BDI-II, ETI, SCL-90-GSI, PSS, PANAS	PTSD associated with MDD (*n* = 17)	Early trauma, severity of perceived stress and general psychopathological symptoms are more closely associated with shorter TL than is the severity of core diagnostic symptoms of PTSD or MDD
Hoen et al. [23]	September 2011	USA	Cohort study	Control	746	68.1 (10.6)	634 (85%)	Smoking (*n* = 131, 18%)	White (85%)	-			CDIS-IV, PHQ-9		Depression is associated with reduced leukocyte telomere length in patients with coronary heart disease but does not predict 5-year change in telomere length
				Case	206	61.7 (10.8)	142 (69%)	Smoking (*n* = 58, 28%)	White (60%)		Antidepressant use (*n* = 99, 48%)	mTL, qPCR, Leukocytes		Major Depression	
AlAhwal et al. [24]	February 2019	Saudi Arabia	Cross-sectional study	Cohort—Patients with colorectal cancer in Saudi Arabia	50	54.5 (11.8)	26 (52%)	-	NA	-	-	TL measurement not specified, rQ-PCR and tissue not specified	SCID-I, HDRS	MDD (10%), disthymia (10%), minor depression (4.0%)	TL progressively shortened from no depressive disorder to minor depression to dysthymia to major depressive disorder. TL was also strongly and inversely correlated with severity of depressive symptoms on the HDRS.
Needham et al. [11]	April 2015	USA	Cross-sectional study	No depression	966	29.2 (5.9)	425 (44%)	NA	Non-Hispanic white (50.2%); African american (19.5%); Mexican American (30.3%)	Anxiety	Antidepressant use (*n* = 32, 3.3%)	RTL, qPCR, Leukocytes	CIDI		Neither depressive nor anxiety disorders were directly associated with telomere length, in young adults. There was suggestive evidence that pharmacologically-treated MD is associated with shorter telomere length, likely reflecting the more severe nature of MD that has come to clinical attention.
				MD or depressed affect	198	30.3 (5.8)	84 (42.4%)		Non-Hispanic white (51.0%); African american (19.7%); Mexican American (29.3%)		Antidepressant use (*n* = 20, 10.1%)			Depressed affect only (*n* = 123, 62.1%); Major depression (*n* = 75, 37.9%)	
Verhoeven et al. [25]	April 2018	USA, Canada and The Netherlands	Cohort study	Year 15	977	40.5 (3.6)	338 (34.6%)	Obese: 32.7%, former smoker: 18.7%, current smoker: 19%; heavy drinker: 9.3%; number of somatic diseases, mean (SD): 0.9 (1.1); physical activity, mean (SD): 336 (274)	White (59.4%)	-	Current antidepressant use (*n* = 67, 6.9%)	mTL, qPCR, Leukocytes	CES-D	CES-D score: 8.5 (7.3); Depression past year (*n* = 86, 8.8%); Depression life-time (*n* = 151, 15.5%)	Evidence for a long-term, between-person relationship of depressive symptoms with LTL, rather than a dynamic and direct within-person relationship
				Year 20		45.4 (3.6)		Obese: 38.5%, former smoker: 21.5%, current smoker: 15.5%; heavy drinker: 10.3%; number of somatic diseases, mean (SD): 1.0 (1.2); physical activity, mean (SD): 334 (273)			Current antidepressant use (*n* = 101, 10.3%)			CES-D score: 9.0 (7.8); Depression past year (*n* = 111, 11.4%); Depression life-time (*n* = 182, 18.6%)	
				Year 25		50.4 (3.6)		Obese: 38.5%, former smoker: 21.5%, current smoker: 15.5%; heavy drinker: 10.3%; number of somatic diseases, mean (SD): 1.0 (1.2); physical activity, mean (SD): 335 (272)			Current antidepressant use (*n* = 109, 11.1%)			CES-D score: 9.0 (7.4); Depression past year (*n* = 131, 13,4%); Depression life-time (*n* = 210, 21.5%)	
Verhoeven et al. [26]	June 2016	Usa and The Netherlands	Cohort study	Baseline	2936	41.8 (13.1)	33.6%	Obese: 16.7%, former smoker: 33.2, current smoker: 38.7%; heavy drinker: 12.7%	NA	Current anxiety disorder (*n* = 535, 32.0%)	Tricyclic antidepressants (*n* = 79, 2.7%); Selective serotonin reuptake inhibitor (*n* = 502, 17.1), Other antidepressants (*n* = 164, 5.6%)	RTL, qPCR, Leukocytes	CIDI	Current Depressive diagnosis disorder (*n* = 389, 23.3%), Control (*n* = 644, 21.9%), Remitted (*n* = 620, 21.1%)	Robust associations of depressive and anxiety disorders with shorter telomeres, but interestingly, it did not demonstrate that depressive and anxiety disorders and LTL change together over time, suggesting the absence of a direct within-person relationship. Short LTL is suggested to be either a long-term consequence or an underlying vulnerability factor for depressive or anxiety disorders
				Year 6	1883	48.6 (12.9)	34.6%	Obese: 19.8%, former smoker: 42.3%, current smoker: 28.2%; heavy drinker: 9.9%		Current anxiety disorder (*n* = 190, 36.0%)	Tricyclic antidepressants (*n* = 56, 3.0%); Selective serotonin reuptake inhibitor (*n* = 224, 11.9), Other antidepressants (*n* = 104, 5.5%)			Current Depressive diagnosis disorder (*n* = 159, 30.1%), Control (*n* = 440, 23.4%), Remitted (*n* = 915, 48.6%)	
Whisman et al. [27]	February 2017	USA	Cross-sectional study		3609	67.0 (10.0)	1404 (43.7)	BMI, mean (SD): 28.4 (6.0), Diabetes (19,5%), Cancer (14.1%), Heart disease (25.0), Stroke (6.0)	White (87.7%), Black (9.7%), Latin (7.4%), Other (2.6%)	-	NA	NA, RT-qPCR, saliva sample	CES-D	Depressive symptoms, mean (SD): 1.5 (2.0)	Higher levels of depressive symptoms were associated with shorter salivary telomeres in men, and this association was incremental to several potential confounds.
Lin et al. [28]	February 2015	USA	Cross-sectional study		464	64.87 (10.99)	368 (79.31)	Current smoker (*n* = 132, 28.45%), former smoker (*n* = 199, 42.89%), Non-Muscle-Invasive Bladder Cancer (*n* = 234, 53.67%)	White (93.1%), Hispanic (3.23%), Black (3.23%), Other (0.44%)	-	NA	RTL, rQ-PCR, Lymphocytes	CES-D, SCID	Alive with CESD < 16 (*n* = 280, 84.08%), Alive with CES-D ≥16 (63, 67.74%), Dead with CES-D < 16 (*n* = 53, 15.92%), Dead with CES-D ≥ 16 (*n* = 30, 32.26%)	Short telomere length and depressive symptoms are associated with bladder cancer mortality individually and jointly.
Zhao et al. [29]	November 2016	USA	Cross-sectional study		2175	40.4 (17.0)	847 (38.9%)	BMI: 31.3 (7.5), smoker (36.4%), drinker (58.0%, physical activity, steps/d, mean (SD): 5845.3 (3899.9)	American indians	-	Antidepressant use: 5.2%	RTL, qPCR, Leukocytes	CES-D	None (50.2%), mild (21.9%), moderate (15.6%), severe (12.3%)	Results showed that individuals with a higher level of depressive symptoms had shorter LTL.
Shaffer et al. [30]	October 2012	USA	Cross-sectional study		2225	48.2 (18.9)	1115 (50.1%)	BMI: 27.0 (5.5), Diabetes mellitus (*n* = 101, 4.5%), Previous ischemic heart disease (*n* = 160, 7.2%)	NA	-	NA	mTL, RT-PCR, Leukocytes	CES-D	Probable depressive disorder (CES-D ≥ 16): (*n* = 269, 12.1%), Elevated depressive symptoms (CES-D ≥ 10): *n* = 613 (27.6%)	Concurrent depressive symptoms were not associated with leukocyte telomere length in a large, representative, population-based study.
Chae et al. [31]	January 2016	USA	Cross-sectional study		92	43.86 (5.73)	92 (100%)	Smorkers (*n* = 51, 55.4%)	African american	Anxiety, mean (SD): 5.02 (3.86)	Medication use (non-specified): (*n* = 29, 31.5%)	mTL, qPCR, Leukocytes	CES-D, HADS	Depression, mean (SD): 7.51 (4.86)	Racial discrimination was associated with shorter LTL among those with lower levels of depressive symptoms
Phillips et al. [32]	April 2013	United Kingdom	Cross-sectional study	Youngest cohort	337	36.6 (0.67)	47%	Smoker (*n* = 76, 23%)	NA	-	Antidepressant use (*n* = 18, 5%)	mTL, qPCR, Leukocytes	HADS	Depression score at wave 5: 3.2 (4.54)	Depressive symptoms are longitudinally associated with shorter telomere length, but only in younger adults.
				Middle cohort	441	57.1 (111)	45%	Smoker (*n* = 111, 25%)			Antidepressant use (*n* = 52, 12%)			Depression score at wave 5: 4.1 (4.66)	
				Older cohort	285	76.1 (0.84)	45%	Smoker (*n* = 38, 13%)			Antidepressant use (*n* = 33, 12%)			Depression score at wave 5: 4.4 (4.50)	
Liu et al. [33]	April 2014	China	Case-control study	Control 1—NGT and no depression	46	51.22 (8.32)	19 (41.30)	Current smokers (*n* = 12, 26.09%); Current drinkers (*n* = 7, 15.22%)	NA	-	NA	RTL, qPCR, Leukocytes	HADS-D	NA	Oxidative stress contributes to both telomere length shortening and depression development in newly diagnosed type 2 diabetic patients, while in depression status, some other mechanisms besides oxidative stress may also affect the telomere length
				Control 2—NGT with depression	6	55.33 (6.56)	3 (50.00)	Current smokers (*n* = 2, 33.33%); Current drinkers (*n* = 1, 16.67%)							
				Case 1—T2DM and no depression	54	54.83 (8.58)	24 (44.44)	Current smokers (*n* = 16, 29.63%); Current drinkers (*n* = 11, 20.37%)							
				Case 2—T2DM with depression	17	54.71 (8.10)	10 (58.82)	Current smokers (*n* = 5, 29.41%); Current drinkers (*n* = 2, 11.76%)							
Wang et al. [34]	April 2017	Sweden	Case-control study	Control	320	44.6 (12.5)	197 (62)	Smokers—NA	NA	-					Telomere length, as compared with healthy controls, is shortened in patients with depression, anxiety and stress and adjustment disorders
				Case	181	41.9 (11.1)	22 (12)	Smokers (*n* = 27, 15%)			Antidepressant use (*n* = 63, 35%)	mTL, rQ-PCR, Leukocyte	PHQ-9, HADS-A/HADS-D, MADRS-S	Baselines—Median score (IQR): MADRS-S—20 (14–25); HAD-D—8 (6–11); HAD-A—12 (9–15); PHQ-9—13 (9–17)	
Schaakxs et al. [35]	June 2014	The Netherlands	Case-control study	Control	128	70.1 (7.2)	49 (38.3)	Cigarette years, median (IQR)—170.00 (598.75); moderate drinker, *n* (%)—99 (77.3); heavy drinker, *n* (%)—12 (9.4);years of education, mean (SD): 12.5 (3.5); Obesity, *n* (%): 26 (20.3%); physical activity, median (IQR): 2.61 (2.96)	NA	-	-	mTL, qPCR, Leukocyte	CIDI, IDS, BAI, AS	-	Mean TL did not differ between depressed persons and never-depressed comparisons
				Case	355	70.6 (7.4)	120 (33,8)	Cigarette years, median (IQR)—100.00 (435.00); moderate drinker, *n* (%)—200 (56.3); heavy drinker, *n* (%)—15 (4.2); years of education, mean (SD): 10.3 (3.4); Obesity, *n* (%): 60 (16.9%); physical activity, median (IQR): 1.60 (2.96)			TCA, *n* (%): 78 (22.0); SSRI, *n* (%): 94 (26.6); Other antidepressants, *n* (%): 101 (28.5); Benzodiazepine use, *n* (%): 141 (39.7)			MDD, *n* (%): 249 (70.1); dysthymia, *n* (%): 6 (1.7); Co-morbid MDD and dysthymia, *n* (%): 92 (25.9%)	
Wolkowitz et al. [36]	May 2011	USA	Case-control study	Control	17	36.6 (11.8)	6 (35%)	BMI: 24.8 (3.7); Daily current tobacco use (6%), Years of education, mean (SD): 15.82 (2.28), Yale Physical Activity Survey, mean (SD): 3.11(0.90)	Caucasian (71%), African-American (18%), Asian (6%) Other or Mixed (5%)		NA	mTL, PCR, Leukocyte	HDRS, SCID, ATHF		The depressed group, as a whole, did not differ from the controls in telomere length. However, telomere length was significantly inversely correlated with lifetime depression exposure, even after controlling for age
				Case	18	36.8 (11.0)	6 (33%)	BMI: 26.3 (5.9); Daily current tobacco use (0%), Years of education, mean (SD): 15.28 (2.06), Yale Physical Activity Survey, mean (SD): 2.10(1.26)	Caucasian (72%), African-American (17%), Asian (6%) Other or Mixed (5%)	-				NA	
Georgin-Lavialle et al. [37]	January 2014	France	Cross-sectional study		19	43.42 (16.97)	4 (21%)	NA	Caucasian (100%)	-	-	mTL, rQ-PCR, Leukocytes	DSM-IV, BDI-II, PSS	-	Telomere length was correlated to perceived stress (r = 0.77; *p* = 0.0001) but not to depression in our population.
Verhoeven et al. [38]	November 2013	USA and The Netherlands	Cohort study	Control	510	40.5 (14.9)	203 (39.8%)	Obesity (13.5%), former smoker (35.6%), current smoker (26.1%), heavy drinker (11.8%), Physical activity (in 1000 MET-minutes per week) (mean ± S.D.): 3.8 (3.0)	NA	-	SSRI (0.4%), benzodiazepine use (0.4%)	mTL, qPCR, Leukocyte	CIDI	IDS score, mean (SD): 5.4 (3.6)	Within the current MDD patients, separate analyses showed that both higher depression severity and longer symptom duration in the past 4 years were associated with shorter TL. The study also confirmed the imprint of past exposure to depression, as those with remitted MDD had shorter TL than controls.
				Case 1: Remitted MDD	802	43.5 (12.5)	238 (29.7%)	Obesity (16.0%), former smoker (26.9%), current smoker (39.8%), heavy drinker (12.3%), Physical activity (in 1000 MET-minutes per week) (mean ± S.D.): 3.8 (3.0)		Comorbid anxiety disorder: 36.9%	TCA (2.9%), SSRI (16.4%), other antidepressant (3.7%), benzodiazepine use (4.4%)			IDS score, mean (SD): 18.0 (10.2)	
				Case 2: Current MDD	1095	40.7 (12.1)	357 (32.6%)	Obesity (20.4%), former smoker (27.8%), current smoker (45.6%), heavy drinker (13.5%), Physical activity (in 1000 MET-minutes per week) (mean ± S.D.): 3.3 (3.1)		Comorbid anxiety disorder: 65.7%	TCA (4.1%), SSRI (29.6%), other antidepressant (11.0%), benzodiazepine use (14.6%)			IDS score, mean (SD): 32.6 (12.2)	
Putermanet et al. [39]	October 2013		Cohort study	Control	743	68.2 (10.5)	84.7%	BMI: 28.29 (5.31), current smoking (17.6%), not at all physical activity (17.4%)	White (60.3%)		Statins (66.8%), Aspirin (72.9%), ARBs and ACEi (52.5%), Antidepressants (10.6%)	RTL, qPCR, Leukocytes	CDIS-IV		MDD was significantly related to LTL at 1 SD below the mean of multisystem resiliency, but not at 1 SD above the mean. This study suggests that MDD associations with biological outcomes should be examined within a psychosocial–behavioral context, because this context shapes the nature of thedirect relationship.
				Case	205	61.7 (10.8)	143(69.8%)	BMI: 29.01 (5.68), current smoking (28.4%), not at all physical activity (22.9%)	White (60.0%)		Statins (58.6%), Aspirin (73.9%), ARBs and ACEi (50.2%), Antidepressants (48.8%)			NA	
Blom et al. [40]	November 2015		Case-control study	Control	63	15.8 (0.2)	22 (35.18%)	NA	NA		NA	STL, qPCR, saliva sample			Adolescents with major depressive disorder exhibited significantly shorter telomere length and significantly smaller right, but not left hippocampal volume.
				Case	54	15.9 (0.2)	29 (53.97%)			Generalized anxiety disorder: 16 Social anxiety disorder: 2 Panic disorder: 1 Specific phobia: 4 Posttraumatic stress disorder: 5 Adjustment disorder: 1 Attention deficit hyperactivity disorder: 8 Alcohol/substance dependence: 1 Conduct disorder: 2 Oppositional defiance disorder: 3 Eating disorder (not otherwise specified): 2			CTQ, BDI-II, CDRS	Beck Depression Inventory II: 26.7 (1.5); Children’s Depression Inventory: 24.1(1.1)	
Vance et al. [41]	April 2018	USA	Cohort study	Control	67	44.1 (14.0)	28 (42%)	Highest educational level: Graduate school (33%), College graduate (42%), Partial college (19%), High school graduate or lower (6%). Living with partner/married: 46%. BMI: 24.3 (3.8). Past year exercise level, more than once a week: (91%). Lifetime alcohol or substance use disorder: (10%). Cigarette smoking pack-years, mean (SD): 8.0 (17.3)	White: 73%; Hispanic/Latino: 3%			RTL, qPCR, Leukocyte	MADRS, HAM-A, PSS, ETISR-SF, TEQ		Individuals with MDD at baseline had greater LTL shortening over two years than individuals without MDD (*p* = 0.03), even after controlling for differences in age, sex, and body mass index (BMI). In the sub-sample of individuals with MDD diagnoses at baseline, no significant associations between LTL change and symptom severity or duration were found.
				Case	50	42.7 (13.2)	24 (48%)	Highest educational level: Graduate school (18%), College graduate (40%), Partial college (28%), High school graduate or lower (18%). Living with partner/married: 18%. BMI: 26.0 (4.7). Past year exercise level, more than once a week: (63%). Lifetime alcohol or substance use disorder: (22%). Cigarette smoking pack-years, mean (SD): 6.8 (12.5)	White: 82%; Hispanic/Latino: 4%		Antidepressant use >6 months: 52%			MDD, *n* (%): 15(37%)	
Starnino et al. [42]	October 2016	Canada	Cohort study		132	45.34 (11.16)	54 (40,9%)	Glasses of alcohol/week, mean (SD): 3.84 (5.38); Smoker n (%): 19(14%), BMI: 25.35 (5.00)	NA	Anxiety and hostility	-	RTL, qPCR, Leukocyte	BDI-II, BAI	Beck Depression Inventory-II, mean (SD): 7.64 (8.21)	Shorter TL was observed among individuals high in defensiveness and depressive symptoms, as well as in those with less hostility and anxiety. Telomere length is associated with psychological burden though the direction of effect differs depending on the psychological variables under study
Wang et al. [43]	September 2019	China	Cross-sectional study		1742	63.6 (4.9)	819 (47.0%)	Education level, *n* (%): Illiterate: 1020 (58.6), Primary: 432 (24.8), Junior: 196 (11.3), Senior or above: 94 (5.4)	NA	-	NA	RTL, q-PCR, Leukocyte	GDS	Mild depressive, *n* (%): 155 (8.9) Severe depressive, *n* (%) 39 (2.2)	Depressive symptoms was negatively correlated with TL in the overall sample. Depressive symptoms significantly mediated the relationship between religiosity and TL (explaining 31.8% of the total variance) in the 65 years and older subgroup
Wikgren et al. [44]	September 2011	Sweden	Case-control study	Control	451	58.9 (11.6)	224 (50%)	BMI: 26.1 (3.5), smoking, *n* (%): 50 (11%)	NA	-	-	NA, qPCR, Leukocytes	BDI, BAI, CES-D, PSQ	CES-D, Median Score (IQR): 6 (3–10)	TL was shorter among patients compared with control subjects (277 base pairs, *p* = 0.001). Overall, short TL was associated with a hypocortisolemic state (low post-DST cortisol and high percentage of cortisol reduction after the DST) among both patients and control subjects but more pronounced among patients.
				Case	91	60.4 (13.1)	36 (40%)	BMI: 26.6 (3.7), smoking, *n* (%): 14 (15%)			Antidepressant use, n (%): 81 (89%)			CES-D, Median Score (IQR): 11 (6–22)	
Szebeni et al. [45]	June 2014	USA	Case-control study	Control	14	51 (5)	13 (93%)	NA	NA	-	NA	RTL, PCR, white matter oligodendrocytes	NA		Relative telomere lengths in white matter oligodendrocytes, but not astrocytes, from both brain regions were significantly shorter for MDD donors as compared to matched control donors
				Case	14	51 (5)	14 (93%)	NA	NA	-	NA			MDD (100%)	
Liu et al. [46]	July 2017	Sweden	Cross-sectional study		894	Age, years, median (IQR): 46 (39, 54)	304 (34%)	Education (%): 0–12 years (51.8%), More than 12 years (48.2%), Obesity (7.4%), Smokers (22.7%), Physical exercise regularity (54.2%)	NA	-	NA	RTL, qPCR, saliva	DSM-IV, AVSI, AVAT	-	In females, depressive status and age had direct negative effects on TL. For males, the effects of stressors and depressive status on TL were mediated by social interaction and the coping strategy worry. In females, no mediation effect of social interaction and coping strategy was detected.
Boeck et al. [47]	June 2018	Germany	Case-control study	Control	21	57.5 (5.7)	0%	BMI: 24.5 (3.0); Smoking, yes, *n* (%): 3(14.3%); Physical activity, yes, *n* (%):18 (85.7%)	NA	-	Medication, *n* (%): Antihypertensive drugs: 3 (14.3%), Tyroid hormone: 3 (14.3%), Sedatives: 1 (4.8%)	mTL, qFISH, PBMC	BDI-II	BDI-II sum score (mean ± S.D.): 2.1 ± 2.2	All of the observed TL changes were signifcantly or marginally signifcantly associated with depressive symptom severity as assessed by the Becks Depression Inventory (BDI-II) sum score. Furthermore, the BDI-II also showed a signifcant negative correlation with TL in memory T helper cells
				Case	18	59.3 (6.6)	0%	BMI: 29.2 (7.5); Smoking, yes, *n* (%): 8(44.4%); Physical activity, yes, *n* (%):11 (61.1%)			Medication, *n* (%): Antidepressants: 13 (72.2%) Antipsychotics: 5 (27.8%) Antihypertensive drugs: 7 (38.9%) Tyroid hormone: 5 (27.8%) Sedatives: 5 (27.8%) Analgesics: 3 (16.7%) Laxatives: 2 (11.1%) Vitamins (B1,B6,B12): 1 (5.6%) Statins: 1 (5.6%)			BDI-II sum score (mean ± S.D.): 23.8 ± 10.9	
Jiménez et al. [48]	October 2018	Colombia	Case-control study	Control	52	21 (3)	F:36 (69%) M:16 (31%)	NA	latin	-	NA	RTL, MMqPCR, Leukocyte	PHQ-9, CES-D, HADS, CTQ	-	Correlation in the clinically significant depressive symptoms group between TL and sexual abuse
				Case	40	21 (3)	F:34 (85%) M:6 (15%)								
Wium-Andersen et al. [49]	January 2017	Denmark	Cross-sectional study	Quartile 1	16,820	52 (42–61)	6943 (41%)	Never smokers, No. (%): 7029 (42), Drinks/week, median (IQR):7 (3–14), Less than 3 years of education, No. (%): 10,062 (60), Low leisure time physical activity, inactive or less than 2–4 h light/day, No. (%): 8658 (51%), BMI, median (IQR): 25 (23–28), chronic disease, No. (%): 5647 (34%)		-	Prescription antidepressant medication, No (%): 7748 (12%)	mTL, modified MMqPCR, Leukocyte	ICD-8, ICD-10	NA	Attendance at hospital for depression was associated with short telomere length cross-sectionally, but not prospectively. Further, purchase of antidepressant medication was not associated with short telomere length cross-sectionally or prospectively. The genetic analyses suggested that telomere length was not causally associated with attendance at hospital for depression or with purchase of antidepressant medication.
				Quartile 2	16,829	55 (45–65)	7391 (44%)	Never smokers, No. (%): 6557 (39), Drinks/week, median (IQR):7 8 (3–15), Less than 3 years of education, No. (%): 10,260 (61), Low leisure time physical activity, inactive or less than 2–4 h light/day, No. (%): 8931 (53), BMI, median (IQR): 25 (23–28), chronic disease, No (%): 6415 (38)							
				Quartile 3	16,828	59 (49–68)	7630 (45%)	Never smokers, No. (%): 6096 (36), Drinks/week, median (IQR):8 (4–15), Less than 3 years of education, No. (%): 10,698 (64), Low leisure time physical activity, inactive or less than 2–4 h light/day, No. (%): 9100 (54), BMI, median (IQR): 26 (23–29), chronic disease, No. (%): 7320 (44)							
				Quartile 4	16,829	64 (54–72)	8100 (48%)	Never smokers, No. (%): 7029 (42), Drinks/week, median (IQR):7 (3–14), Less than 3 years of education, No. (%): 10,062 (60), Low leisure time physical activity, inactive or less than 2–4 h light/day, No. (%): 8658 (51%), BMI, median (IQR): 25 (23–28), chronic disease, No (%): 5647 (34%)	NA						
Huzen et al. [50]	January 2010	The Netherlands and United Kingdom	Cross-sectional study		890	73 (64–79)	535 (61%)	BMI: 26 (24–30)	NA	-	-	mTL, RT-qPCR, Leukocyte	CES-D, DS14, RAND-36	Severe depression, *n* (%): 154 (18%), depressive symptoms only, *n* (%): 145 (16,3%)	A lower perceived mental health on the RAND-36 score was associated with shorter telomere length. Telomere length was not associated with the CES-D or DS14 score.
Simon et al. [51]	August 2015	USA	Case-control study	Control	166	41.3 (13.7)	77 (46%)	Educational level: Graduate school: 35%, College graduate (40%), Partial college (18%), High school graduate or less (7%). Lifetime alcohol or drug abuse/dependence, N (%): 13 (8). BMI: 25.4 (4.2)	White: 68%, Black or African American:17%, Asian:7%, Native American/Alaska Native: 0%, Other: 8%. Not Hispanic/Latino: 90%, Hispanic/Latino: 10%			RTL, RT-qPCR, Southern blot, Leukocytes	DSM-IV, SCID, MADRS, CIRS, TEQ, ETISR-SF, ICG	MADRS Total Score, Mean (SD): 1.9 (2.4)	Our well-characterized, well-powered examination of concurrently assessed telomere length and telomerase activity in individuals with clinically significant, chronic MDD and matched controls failed to provide strong evidence of an association of MDD with shorter LTL, while telomerase activity was lower in men with MDD.
				Case	166	41.3 (13.8)	77(46%)	Educational level: Graduate school: 20%, College graduate (35%), Partial college (25%), High school graduate or less (17%). Lifetime alcohol or drug abuse/dependence, *N* (%): 47 (28%)	White: 78%, Black or African American:11%, Asian:4%, Native American/Alaska Native: 1%, Other: 4%. Not Hispanic/Latino: 92%, Hispanic/Latino: 7%	Current anxiety disorder (50%), litime anxiety disorder (55%)	Anti-depressant use > 6 months, *N* (%): 65 (39), Mood stabilizer use > 6 months, *N* (%): 6 (4), Benzodiazepine use > 6 months, *N* (%): 19 (11) Antipsychotic use > 6 months, *N* (%): 10 (6)			MADRS Total Score, Mean (SD): 28.2 (6.0)	
Harttman et al. [52]	November 2010	Germany	Case-control study	Control	20	49.1 (15.2)	11 (55%)	Smokers (*n* = 7, 35%)	NA	-					Major depressive disorder is associated with shortened telomeres. However, differences in the applied therapy, the duration of illness, or the severity of depression do not seem to have any influence on telomere length.
				Case	54	49.1 (14.1)	21 (39%)	Smokers (*n* = 16, 29.6%)	NA	-	TAD ≤ 1: *n* = 20, TAD > 1 *n* = 16	mTL, Southern blot, Leukocyte	HAM-D	HAM-D, (SD) [Range]: 29.9 (6.0) [7,8,9,10,11,12,13,14,15,16,17,18,19,20,21,22,23,24,25,26,27,28,29,30,31,32,33,34,35,36]	
Karabatsiakis et al. [53]	July 2014	Germany	Case-control study	Control	50	51.1 (8)	0%	Years of education, mean (SD): 15.1 (2.4)	NA	-		mTL, qFISH, Leukocyte	BDI	NA	: A history of depression is associated with shortened telomeres in the main effector populations of the adaptive immune system. Shorter telomeres seem to persist in individuals with lifetime depression independently of the severity of depressive symptoms.
				Case 1—Lifetime depressed with irrelevant symptoms	24	53.1 (7.2)	0%	Years of education, mean (SD): 14.1 (2.1)			SSRI/ SNRI intake, n (%): 10 (41.7%)				
				Case 2—Relevant symptoms of depression	20	53.8 (7.6)	0%	Years of education, mean (SD): 14.2 (2.9)			SSRI/ SNRI intake, n (%): 9 (45%)				
Solomon et al. [54]	July 2017	Israel	Cohort study		99	63.6 (3.7)	NA	Years of education (M, SD): 14, 3.7, Physical exercising regularly: 62 (62.6%) Smoking on a regular basis: 21 (21.2%)	NA	PTSD	NA	mTL, Southern blot, Leukocyte	SCL-90	NA	Ex-POWs had shorter telomeres compared to controls (Cohen’s d = 0.5 indicating intermediate effect). Ex-POWs with chronic depression had shorter telomeres compared to those with delayed onset of depression (Cohen’s d = 4.89), and resilient ex-POWs (Cohen’s d = 3.87), indicating high effect sizes.
Verhoeven et al. [55]	November 2019	The Netherlands	Cohort study		2032	42.5 (12.8)	665 (32.7%)	NA	European ancestry: 2032 (100%)	-	NA	RTL, qPCR, Leukocyte	CIDI, DSM-IV, IDS-SR, NEO-FFI	Lifetime depression diagnosis (% yes, N): 83.1, 1688 Depression severity (average 6-year IDS score) (mean ± S.D.): 19.0 (12.0)	The use of genetic methods in this paper indicated that the established phenotypic association between telomere length and depression is unlikely due to shared underlying genetic vulnerability. These findings suggest that short telomeres in depressed patients may simply represent a generic marker of disease or may originate from non-genetic environmental factors
Mamdani et al. [56]	September 2015	USA	Case-control study	Control	10	48 (13.0)	7 (70%)	NA	NA	Schizofrenia and bipolar disorder	NA	RTL, qPCR, brain tissue	UCCIB psychological autopsy protocol		A significant decrease in telomere length was observed specifically in the HIPP of MDD subjects even after controlling for age. In the HIPP of MDD subjects, several genes involved in neuroprotection and in stress response showed altered levels of mRNA.
				Case	10	47.3 (11.5)	3 (30%)	NA	NA					MDD: 10 (100%)	

List of abbreviations: mTL—mean telomere length; RTL—relative telomere length; PCR—Polymerase Chain Reaction; qPCR—Quantitative PCR; RT-PCR—real-time PCR; rQ-PCR or RT-qPCR—real-time quantitative PCR; MMqPCR—monochrome multiplex quantitative PCR; qFISH—Quantitative Fluorescent in situ hybridization; T2DM—Type 2 diabetes mellitus; NGT—normal glucose tolerant; mtDNA—mitochondrial DNA; TCA—tricyclic antidepressants; SSRI—selective serotonin reuptake inhibitor; ARBs—Angiotensin II receptor blockers; ACEi—Angiotensin Converting Enzyme inhibitors; TAD—Total antidepressant dose; HIPP—hippocampus. POW—prisoners of war; ACE—adverse childhood experiences: DST—dexamethasone suppression test; BMI—Body Mass Index—Mean, (SD).

**Table 2 cells-10-01423-t002:** Study characteristics, clinical and epidemiological data analysis of PTSD impacts on telomere lenght.

Authors	Date of Publication	Country	Study Design		Patients	Age, Years, Mean (SD)	Sex, *n* (%) Male	Main Comorbidities/ Lifestyle Factors Associated	Race/Ethnicity	Other Associated Psychiatric Diseases in this Study	Medication	Telomere Measurement and Tissue	Measurement of Psychiatric Disorder	Telomere Lenght
Zhang et al. [57]	April 2019	USA	Cross-sectional	Low hostility	135	31.3 ± 8.7	127 (29.8%)	*n* (%) <12th grade 3 (2.2), High school diploma or G.E.D 33 (24.6), Some college/technical school 57 (42.5), Bachelor’s degree 24 (17.9), Graduate degree 17 (12.7)	364 Whites, 61 Blacks, 25 Asian or Pacific Islanders, 11 American Indian or Alaskan Natives, and 13 unknown	Depression (28%) and suicide ideation (24%) of the PTSD subjects	NA	RTL, qPCR, Leukocytes	BSI, PCL	Among the participants with PTSD, those with medium or high level of hostility had shorter LTL than those with low level hostility (*p* < 0.01). Stepwise regression indicated that hostility level and age, but not gender and PTSD, were negatively correlated with LTL.
				Medium hostility	267	28.5 ± 7.1	237 (55.6%)	<12th grade 1 (1.1), High school diploma or G.E.D 73 (27.2), Some college/technical school 133 (49.6), Bachelor’s degree 46 (17.2), Graduate degree 13 (4.8)						
				High hostility	72	26.5 ± 5.9	62 (14.6%)	<12th grade 3 (1.4), High school diploma or G.E.D 37 (51.4), Some college/technical school 29 (40.3), Bachelor’s degree 5 (6.9), Graduate degree 0 (0.0)						
Bersani et al. [22]	October 2015	USA	Cross-sectional	Control	41	34.64 (9.17)	41 (100%)	Years of education (mean ± SD): 14.79 ± 2.44, current smokers (*n*): 11	35 hispanic and 42 non-hispanic	Depression	Statins (*n* = 2), NSAIDs (*n* = 5), antidepressants (*n* = 13), antibiotics (*n* = 1), hormone drugs for prostate cancer (*n* = 1), analgesics (*n* = 1)	RTL, PCR, Granulocytes	CAPS, BDI-II, ETI, SCL-90-GSI, PSS, PANAS	TL was negatively correlated with early trauma (*p* < 0.001),global psycho-pathological severity (*p* = 0.044) and perceived stress (*p* = 0.019), positively correlated with positive affect (*p* = 0.026), not significantly correlated with symptom severity of PTSD, depression or negative affect.
				Case	35 (17 also w/ MDD)		35 (100%)							
O’Donovan et al. [58]	September 2011	USA	Cross-sectional	Control	47	30.68 ± 8.19	21 (44.68%)	Female: Education: 15.4 (2.0), BMI: 25.2 (4.2), Current smoker *n* (%): 5 (20), Alcohol use *n* (%): 0 (0), Substance use *n* (%): 1 (4) Past abuse.	6 African americans, 9 ssian americans, 57 whites, 1 hispanic, 2 hawaiians, 4 pacific islanders and 9 multi-ethnics	23 past MDD, 8 current MDD	-	qPCR	CAPS, DSM-IV	Participants with PTSD had shorter age-adjusted LTL than controls. Exposure to childhood trauma was also associated with short LTL. In fact, childhood trauma appeared to account for the PTSD group difference in LTL; only participants with PTSD and exposure to multiple categories of childhood trauma had significantly shorter LTL than controls.
								Male: Education: 15.5 (2.1), BMI: 23.6 (3.1), Current smoker *n* (%): 3 (14), Alcohol use *n* (%): 1 (5) Past abuse, Substance use *n* (%): 0 (0).						
				Case	43	30.60 ± 6.63	23 (53.48%)	Female: Education: 15.2 (2.1), BMI: 23.9 (2.0), Current smoker *n* (%): 6 (30), Alcohol use *n* (%): 4 (20 Past abuse and 4 (20) past dependence, Substance use *n* (%): 1 (5) past abuse and 2(10) past dependence						
								Male: Education: 14.4 (2.3), BMI: 29.5 (4.3), Current smoker *n* (%): 4 (18), Alcohol use *n* (%): 4 (18) Past abuse and 4 (18) past dependence, Substance use *n* (%): 0 (0) Past abuse and 4 (18) past dependence						
Kang et al. [59]	July 2020	USA	Cross-sectional	Control 1: low combat exposure	59	33.17 (8.69)	83 (100%)	Education, Years: 15.01 ± 2.09, Body Mass Index, kg/m2: 28.19 ± 4.14, Waist Circumference, cm: 93.47 ± 11.14, Smoker, Yes/No, *n*: 17/94, Alcohol Abuse, Yes/No, *n*: 8/103, Substance Use, Yes/No, *n*: 2/109, MDD Diagnosis by SCID, Yes/No, *n*: 0/111, Time Since Severe Combat Event, Months: 65.85 ± 35.15	31 Hispanic, 80 others	-	3 Antidepressant, 6 benzodiazepines and hypnotics and 1 anticonvulsants	RTL, qPCR, Leukocytes	DSM-IV, CAPS, PSS, PANAS	Subjects with PTSD showed shorter telomere length and larger amygdala volume than those without PTSD among veterans exposed to high trauma, while there was no significant group difference in these parameters among those exposed to low trauma.
				Control 2: high combat exposure	24									
				Case 1: low combat exposure	12	33.66 (8.17)	65 (100%)	Education, Years: 13.93 ± 1.90, Body Mass Index, kg/m^2^: 29.79 ± 5.58, Waist Circumference, cm: 98.61 ± 14.72, Smoker, Yes/No, *n*: 33/69, Alcohol Abuse, Yes/No, *n*: 13/89, Substance Use Yes/No, *n*: 6/96, MDD Diagnosis by SCID, Yes/No, *n*: 50/52, Time Since Severe Combat Event, Months: 77.67 ± 30.95	46 Hispanic, 56 others	50 MDD	21 Antidepressant, 8 benzodiazepines and hypnotics, 2 antipsychotics and 5 anticonvulsants			
				Case 2: high combat exposure	53									
Malan et al. [60]	August 2011	South Africa	Cohort	Control	53	22.3	0 (0)	High school education (Grade 8 and higher) (59 (92%)), primary school education (5 (7.81%))	12 (19%) black, 1 (1%) white, and 51 (80%) individuals of mixed ancestry.	23 (36%) diagnosed with MDD at baseline and 31 (48%) diagnosed with MDD at the 3-month follow-up.	NA	RTL, qPCR, Leukocytes	DSM-IV, CD-RISC, BDI, CES-D, MADRS, ETI	A marginally significant association was evident between relative LTL and PTSD status.
				Case	9		0 (0)				NA			
Boks et al. [61]	January 2015	Netherlands	Cross-sectional	Control: low trauma	0	25.1 (8.1)	128 (100%)	Smokers = 19, Increase alcohol use = 15, Decrease alcohol use = 10, Unchanged alcohol use = 31	Dutch ethinicity	-	antibiotics in 4 cases, antihistamines in 3 cases and one case of a benzodiazepine prescription. Stopped medication included: antibiotics in one case, antihistaminic in two cases and one case of oral isotretinoïn.	mTL, qPCR, blood sample	SRIP, ETI	Development of post traumatic stress disorder (PTSD) symptoms was significantly associated with increased telomere length and decreased DNAm ageing.
				Case: low trauma	64									
				Control: high trauma	32	27.4 (9.3)								
				Case: high trauma	32									
Roberts et al. [62]	May 2017	USA	Cross-sectional	Control	25	45.5 (3.6)	0 (0)	BMI, blood draw, kg/m2 (Mean (SD)): 25.1 (4.0), Past-month smoking, blood draw, any %(*N*): 14.3 (4), Past-month alcohol consumption, blood draw, none %(*N*): 32.1 (9), Diet, least healthy quintile, 1995 %(*N*): 12.0 (3), Past-month physical activity, blood draw, <1/week %(*N*): 32.1 (9), High cholesterol, 1995 %(*N*): 7.1 (1), High blood pressure, 1995 %(*N*): 3.6 (1)	NA	Depression	Antidepressant 10.7%	RTL, RT-qPCR, Leukocytes	DSM-IV, PHQ-9, PCL-C	Relative to not having PTSD, women with a PTSD diagnosis had shorter log-transformed TL. Adjustment for health behaviors and medical conditions did not attenuate this association. Trauma type was not associated with TL and did not account for the association of PTSD with TL.
				Case	66 subclinical PTSD	46.6 (3.7)	0 (0)	BMI, blood draw, kg/m2 (Mean (SD)): 26.0 (7.3), Past-month smoking, blood draw, any %(N): 9.0 (6), Past-month alcohol consumption, blood draw, none %(*N*): 43.9 (29), Diet, least healthy quintile, 1995 %(*N*): 15.2 (10), Past-month physical activity, blood draw, <1/week %(*N*): 39.4 (26), High cholesterol, 1995 %(*N*): 7.6 (5), High blood pressure, 1995 %(*N*): 3.0 (2)			Antidepressant 9.1%			
					25 PTSD diagnosis	46.6 (3.9)	0 (0)	BMI, blood draw, kg/m2 (Mean (SD)): 28.4 (7.6), Past-month smoking, blood draw, any %(*N*): 16.0 (4), Past-month alcohol consumption, blood draw, none %(*N*): 56.0 (14), Diet, least healthy quintile, 1995 %(*N*): 28.0 (7), Past-month physical activity, blood draw, <1/week %(*N*): 32.0 (8), High cholesterol, 1995 %(*N*): 12.0 (3), High blood pressure, 1995 %(*N*): 8.0 (2)			Antidepressant 52%			
Ladwig et al. [63]	July 2013	Germany	Cross-sectional	Control	2687	56.5	1330 (49.5%)	Means (SD) Low educational level (%) 59.0; Living alone (%) 24.1; BMI (kg/m2) 27.6 (4.8); Current smoking (%) 17.8; Alcohol consumption: No 30.1, Moderate 52.7, High 17.2; Physical inactivity (%) 45.7; Actual hypertension (%) 31.7; TC/HDL-C 4.09 (1.18); History of chronic diseases 16.8	NA	Depression (PHQ-9) (%)* 3.9, Depressed mood/exhaustion (DEEX) (%)* 18.4	NA	mTL, qPCR, Leukocytes	PDS, PHQ-9, DEEX	The multiple model revealed a significant association between partial PTSD and TL as well as between full PTSD and shorter TL indicating shorter TL on average for partial and full PTSD. An additional adjustment for depression and depressed mood/exhaustion gave comparable beta estimations.
				Case	262 partial PTSD	52.5	100 (38.2%)	Low educational level (%) 57.6; Living alone (%) 29.4; BMI (kg/m^2^) 27.4 (5.1); Current smoking (%) 21.4; Alcohol consumption: No 30.2, Moderate 49.6, High 20.2; Physical inactivity (%) 42.7; Actual hypertension (%) 24.4; TC/HDL-C 4.04 (1.20); History of chronic diseases 22.9		Depression (PHQ-9) (%)* 13.0, Depressed mood/exhaustion (DEEX) (%)* 55.0				
					51 full PTSD	54.5	19 (37.3%)	Low educational level (%) 64.7; Living alone (%) 35.3; BMI (kg/m2) 28.1 (5.7); Current smoking (%) 9.8; Alcohol consumption (%): No 35.3, Moderate 41.2, High 23.5; Physical inactivity (%) 47.1; Actual hypertension (%) 27.5; TC/HDL-C 4.02 (1.21); History of chronic diseases 19.6		Depression (PHQ-9) (%)* 5.9, Depressed mood/exhaustion (DEEX) (%)* 56.9				
Jergović et al. [64]	June 2014	Croatia	Case-control study	Control	17 age-matched	47.2 (1.71)	17 (100%)	Body mass index 27.13 ± 4; Education: Elementary/high 29 (96.4), University 1 (3.4) Work status: Employed 1 (3.4); Unemployed/retired 29 (96.4); Tobacco use: Yes 17 (58.6); No 12 (41.4); Alcohol use: Yes 5 (20); No 24 (80); Daily physical exercise 1 (3.33)	NA	-	NA	RTL, RT-PCR, PBMCs	CAPS, STAI, BDI	Middle-aged war veterans with current PTSD had shorter PBMC telomere length than their age-matched healthy controls while the elderly had the shortest telomeres.
					15 elderly	80 or older	2 (13.33%)	NA	NA	-	NA			
				Case	30	45.9 (1.12)	30 (100%)	mean ± SD Body mass index 27.3 ± 2.62; Education: Elementary/high 29 (96.4), University 6 (35.3); Work status: Employed 15 (88.2), Unemployed/retired 2 (11.8); Tobacco use: Yes 9 (52.9), No 8 (47.1); Alcohol use: Yes 11 (64.7), No 6 (35.3); Daily physical exercise 1 (5.88)	NA	24 (80%) MDD, 13 (43%) panic disorder, 9 (30%) obsessive compulsive disorder, 7 (23%) social phobia	Analgesics (non-steroidal anti-inflammatory drugs, *N* = 18, 60%; opioid analgesics, *N* = 3, 10%), hypolipidemics (*N* = 3, 10%), antihypertensives (*N* = 3, 10%), proton pump inhibitor (*N* = 1, 3%), (*N* = 28, 93%) were treated with psychotropic medication: antidepressants (*N* = 27, 90%), mood stabilizers (*N* = 7, 23%), anxiolytics (*N* = 26, 87%), hypnotics (*N* = 22, 73%), and antipsychotics (*N* =14, 47%).			
Avetyan et al. [65]	April 2019	Armenia	Case-control study	Control	49	43.5 (9.4)	49 (100%)	NA	NA	-	NA	RLT, qPCR, Leukocytes	SCID-I, CAPS	Comparison of LTL in diseased and healthy subjects showed that PTSD patients had 1.5 times shorter average LTL than controls.
				Case	41	46.4 (7.63)	41 (100%)	NA						
Kim et al. [66]	June 2017	South Korea	Cross-sectional	Control —High combat exposure	11	62.82 (5.74)	11 (100%)	Education (years) 9.45 (4.28) Socioeconomic status: High/Medium/Low, *n* 2/5/4 AUDIT score 6.73 (7.56) Heavy smoker: Yes/No, *n* 8/3	NA	-	20.8% psychoactive medications	RLT, qPCR, Leukocytes	CAPS, CES, AUDIT	As a whole, no significant difference in telomere length was found between PTSD and non-PTSD groups. In linear regression analysis stratified by trauma levels, among veterans exposed to severe combat, PTSD status, antidepressant use, and education level affected telomere length.
				light-to-moderate combat exposure	109	62.95 (4.23)	109 (100%)	Education (years) 10.56 (3.03) Socioeconomic status: High/Medium/Low, *n* 18/50/41 AUDIT score 6.71 (7.67) Heavy smoker: Yes/No, *n* 65/44						
				Case—High combat exposure	34	63.38 (3.13)	34 (100%)	Education (years) 10.35 (3.28) Socioeconomic status: High/Medium/Low, *n* 7/14/13 AUDIT score 13.09 (10.77) Heavy smoker: Yes/No, *n* 20/14			71.3% psychoactive medications			
				light-to-moderate combat exposure	88	62.84 (3.50)	88 (100%)	Education (years) 10.38 (2.63) Socioeconomic status: High/Medium/Low, *n* 18/41/29 AUDIT score 11.01 (10.97) Heavy smoker: Yes/No, *n* 41/47						
Solomon et al. [54]	July 2017	Israel	Cohort	Control: Resilient	47	63.6 (3.7)	NA	Years of education (M, SD): 14, 3.7, Physical exercising regularly: 62 (62.6%) Smoking on a regular basis: 21 (21.2%)	NA	Depression	NA	mTL, Southern blot, Leukocytes	PTSD-I, DSM-IV	PTSD trajectories were not implicated in telomere length.
				Case 1: chronic PTSD	5									
				Case 2: delayed PTSD	46									
				Case 3: recovered PTSD	1									
Zhang et al. [67]	November 2013	USA	Cross-sectional	Control	566	29.2 ± 7.3	412 (72.79%)	NA	65.3% were White, 13.9% were Black, 7.8% were Asian or Pacific Islander, and 12.9% were American Indian or Alaskan Native	-	NA	RTL, RT-PCR, Leukocytes	PCL, DSMI-IV, SLE	Participants with PTSD had a lower relative T/S ratio than non-PTSD control subjects. This remained true when PTSD subjects were compared with age-matched non-PTSD controls.
				Case	84		76 (90.47%)	NA	75.0% were White, 10.2% were Black, 4.5% were Asian or Pacific Islander, and 10.2% were American Indian or Alaskan Native					

List of abbreviations: mTL—mean telomere length; RTL—relative telomere length; PCR—Polymerase Chain Reaction; qPCR—Quantitative PCR; RT-PCR—real-time PCR; rQ-PCR or RT-qPCR—real-time quantitative PCR; TL—telomere length; LTL—leukocyte telomere length; PBMC—peripheral blood mononuclear cell; BMI—body mass index, mean (SD); TC/HDL-C—total cholesterol/ high-density lipoprotein-cholesterol; NSAID—nonsteroidal anti-inflammatory drug; MDD—major depressive disorder; G.E.D—General Educational Development.

**Table 3 cells-10-01423-t003:** Study characteristics, clinical and epidemiological data analysis of anxiety disorders impacts on telomere lenght.

	Date of Publication	Country	Study Design		Patients	Age, Years, Mean (SD)	Sex, *n* (%) Male	Main Comorbidities/ Lifestyle Factors Associated	Race/Ethnicity	Other Associated Psychiatric Diseases in This Study	Medication	Telomere Measurement and Tissue	Measurement of Psychiatric Disorder	Level of Anxiety	Telomere Lenght
Verhoeven et al. [68]	January 2018	The Netherlands	Case control study	Control	582	41.7 (14.8)	230 (39.5%)	Obesity (13.2%), former smoker (35.6%), current smoker (26.8%), moderate drinker (78.2%), heavy drinker (11.5). Years of education, mean (SD): 12.9 (3.2)	NA	Agoraphobia, panic disorder, socialphobia, generalised anxiety disorder	Antidepressant use, % Tricyclic antidepressant (0.2) Selective serotonin reuptake inhibitor (0.5), other antidepressant (0.2). Benzodiazepine use, %: (0.5)	mTL, qPCR, Leukocytes	BAI	BAI, mean (SD): 2.9 (2.9)	Patients with current—but not remitted—anxiety disorder had shorter telomere length, suggesting a process of accelerated cellular ageing, which in part may be reversible after remission.
				Case 1—Remitted anxiety	459	43.6 (12.7)	131 (28.5%)	Obesity (17%), former smoker (39.8%), current smoker (33.8%), moderate drinker (73.2%), heavy drinker (11.8%). Years of education, mean (SD): 12.4 (3.4)			Antidepressant use, % Tricyclic antidepressant (2.4) Selective serotonin reuptake inhibitor (16.3), other antidepressant (3.5). Benzodiazepine use, %: (3.9)			BAI, mean (SD): 8.9 (7.3)	
				Case 2—Current anxiety group	1283	41.3 (12.4)	412(32.1%)	Obesity (18.4%), former smoker (29.6%), current smoker (45.6%), moderate drinker (65%), heavy drinker (13.1%). Years of education, mean (SD): 11.6 (3.3)			Antidepressant use, % Tricyclic antidepressant (4.4) Selective serotonin reuptake inhibitor (26.8), other antidepressant (9.1). Benzodiazepine use, %: (13.1)			BAI, mean (SD): 18.5 (10.8)	
Groer M et al. [69]	December 2019	USA	Cross-sectional		97	29.6 (6.3)	0	Smoking (*n* = 2), more than an hour of exercise per week (*n* = 5), BMI: 28.8 (5.9). Completed completion or postgraduate education preparation (54%)	The sample was 76% Caucasian (39% of whom were of Hispanic origin) and 15% African American. The remaining 9% were Asian or other racial categories	Depression	NA	RTL, qPCR, DNA was extracted from PBMCs with DNeasy Blood and Tissue Ki	POMS, PSS	NA	There were no statistically significant relationships between TL and demographic factors, stress, depression, or TPO status. There were significant negative correlations between TL and anxiety and a trend for a relationship between TL and IL-6 levels. IL-6 levels were significantly, positively associated with negative moods.Higher anxiety scores and inflammation were associated with shorter TL. Inflammation was related to anxiety and other dysphoric moods and was marginally associated with shorter TLs.
Schaakxs et al. [71]	April 2015	The Netherlands	cross-sectional study	Control	128	70.1 (7.2)	49 (38.3)	Cigarette years, median: 170.00 (598.75); moderate drinker, *n* (%): 99 (77.3); heavy drinker, *n* (%): 12 (9.4);years of education, mean (SD): 12.5 (3.5); Obesity, *n* (%): 26 (20.3%); physical activity, median (IQR): 2.61 (2.96)	NA	MDD, *n* (%): 249 (70.1); dysthymia, *n* (%): 6 (1.7); Co-morbid MDD and dysthymia, *n* (%): 92 (25.9%)	-	mTL, qPCR, Leukocyte	CIDI, IDS, BAI, AS, CTI	NA	Mean TL did not differ between depressed persons (bp (SD): 5035 (431)) and never-depressed (bp (SD): 5057 (729)) comparisons. Further, TL was not associated with severity, duration, and age at onset of depression; comorbid anxiety disorders; anxiety symptoms; apathy severity; antidepressant use; benzodiazepine use; cognitive functioning; and childhood trauma
				Case	355	70.6 (7.4)	120 (33.8)	Cigarette years, median (IQR): 100.00 (435.00); moderate drinker, *n* (%): 200 (56.3); heavy drinker, *n* (%): 15 (4.2); years of education, mean (SD): 10.3 (3.4); Obesity, *n* (%): 60 (16.9%); physical activity, median (IQR): 1.60 (2.96)			TCA, *n* (%): 78 (22.0); SSRI, *n* (%): 94 (26.6); Other antidepressants, *n* (%): 101 (28.5); Benzodiazepine use, *n* (%): 141 (39.7)				
Wang et al. [34]	April 2017	Sweden	Case-control study	Control	320	44.6 (12.5)	197 (61.5%)	Smokers—NA	NA	Depression and stress and adjustment disorders		mTL, qRT-PCR, Leukocyte	PHQ-9, HADS-A/HADS-D, MADRS-S		Telomere length, as compared with healthy controls, is shortened in patients with depression, anxiety and stress and adjustment disorders
				Case	181	41.9 (11.1)	22 (12.15%)	Smokers (*n* = 27, 15%)			Antidepressant use (*n* = 63, 35%)			Baselines—Median score (IQR): MADRS-S: 20 (14–25); HAD-D: 8 (6–11); HAD-A: 12 (9–15); PHQ-9: 13 (9–17)	
Tyrka et al. [18]	January 2016	USA	Case-Control study	Control	113	28.5 (9.2)	50 (44.2%)	Smokers (8.3%)	White (82.3%)	Adversities, depression, PTSD	NA	mTL, qPCR,, Leukocytes	SCID, STAI, PSS, CD-RISC		Significantly higher mtDNA copy numbers and shorter telomeres were seen in individuals with major depression, depressive disorders, and anxiety disorders, as well as those with parental loss and childhood maltreatment.
				Case 1—Adverity with no psychiatric disorder	66	31.3 (11.1)	26 (39.4%)	Smokers (7.8%)	White (80.3%)						
				Case 2—Psichyatric disorder with no adversity	39	30.7 (10.4)	15 (38.5%)	Smokers (7.7%)	White (92.3%)					MDD (*n* = 6), depressive (*n* = 7)	
				Case 3—Adversity and psychiatric disorder	72	34.8 (12.0)	22 (30.6%)	Smokers (17.1%)	White (81.9%)					MDD (*n* = 7), depressive (*n* = 18)	
Prelog M et al. [70]	June 2016	Germany	Case-control study	Control	129	Female (*n* = 85): 36.8 (10.9) Male (*n* = 44): 34.1(10.8)	44 (34.1%)	Not documented	NA	Depression (*n* = 46)		RTL, RT-qPCR, Leukocytes	SCID-I	Panic disorder (*n* = 129)	Relative telomere lengths (RTLs) were not different between patients and HC. However, within the patient group, smokers had significantly shorter telomeres (0.91 ± 0.30) compared to non-smokers (1.07 ± 0.37) (*p* = 0.018) and females (0.96 ± 0.34) had shorter telomeres than males (1.10 ± 0.32)
				Case	129	Female (*n* = 85): 36.9 (10.8) Male (*n* = 44): 34.1(11.7)	44 (34.1%)	Smokers (*n* = 39)			Antidepressants, yes, *n* (%): 52 (40.3%)				
B L Needham et al. [11]	September 2014	USA	Cross-sectional study	Control—No anxiety	952	29.3(5.8)	413 (43.4%)	NA	Non-Hispanic white 485 (51%); African american 184 (19.3%); Mexican American 283 (29.7%)	-	Antidepressant use, *n* (%): 27 (2.8%)	mTL, qPCR, Leukocytes	CIDI	-	The primary finding from this study is that depressive and anxious symptomatology, overall, have no direct relationship with TL in young adulthood. Although associations did not vary by race/ethnicity, among women (but not men) past-year GAD/PD was associated with shorter TL. There was no direct effect of antidepressant medication use on TL, but among current users of antidepressants, those with past-year MD had shorter TL than those with no depression. To our knowledge, this is the first study to examine relationships between antidepressant medication use and depressive and anxious symptomology, as well as the first to examine variation in these relationships by race/ethnicity, in a nationally representative sample.
				Case 2—GAD/PD or anxious affect	212	30.0(6.0)	94 (44.3%)		Non-Hispanic white 101 (47.6%); African american 43 (20.3%); Mexican American 68 (32.1%)		Antidepressant use, *n*(%): 25 (11.8%)				
Hoen et al. [21]	August 2012	Netherlands	Longitudinal study	Control	980	53.7 (11.3)	F: 551 (56.22%) M: 465 (43.78%)	Smoking (*n* = 225; 77%), Alcohol consumption (*n* = 788; 80%), Sedentarism (*n* = 50; 52%)	NA	Depression		mTL, PCR, Leukocytes	CIDI	NA	No association was found between depressive disorders and shorter telomeres at follow-up. Anxiety disorders predicted shorter telomere length at follow-up in a general population cohort.
				Case	97	51.3 (10.7)	F: 62 M: 36	Smokers (*n* = 32; 65%); Alcohol consumption (*n* = 78; 80%), Sedentarism (*n* = 505; 52%)			Antidepressant use (*n* = 14; 15%)				
Verhoeven et al. [26]	June 2016	USA and The Netherlands	Longitudinal study	Baseline	2936	41.8 (13.1)	986 (33.6%)	Former Smokers (*n* = 975), Current Smokers (*n* = 1.136); Mild-moderate drinker (*n* = 2064), Heavy drinker (*n* = 373)	NA	Current Depressive diagnosis disorder (*n* = 389, 23.3%), Control (*n* = 644, 21.9%), Remitted (*n* = 620, 21.1%)	Tricyclic antidepressants (*n* = 79, 2.7%); Selective serotonin reuptake inhibitor (*n* = 502, 17.1), Other antidepressants (*n* = 164, 5.6%)	RTL, qPCR, Leukocytes	CIDI	Current anxiety disorder (*n* = 535, 32.0%)	Robust associations of depressive and anxiety disorders with shorter telomeres, but interestingly, it did not demonstrate that depressive and anxiety disorders and LTL change together over time, suggesting the absence of a direct within-person relationship. Short LTL is suggested to be either a long-term consequence or an underlying vulnerability factor for depressive or anxiety disorders
				Year 6	1883	48.6 (12.9)	641 (34.6%)	Former Smokers (*n* = 797), Current Smokers (*n* = 531); Mild-moderate drinker (*n* = 1367), Heavy drinker (*n* = 186)		Current Depressive diagnosis disorder (*n* = 159, 30.1%), Control (*n* = 440, 23.4%), Remitted (*n* = 915, 48.6%)	Tricyclic antidepressants (*n* = 56, 3.0%); Selective serotonin reuptake inhibitor (*n* = 224, 11.9), Other antidepressants (*n* = 104, 5.5%)			Current anxiety disorder (*n* = 190, 36.0%)	
Chae et al. [31]	September 2015	USA	Cross-sectional study		92	43.86 (5.73)	92 (100%)	Education, *n* (%): high school or less 38 (41.3), some college or more 54 (58.7). Work status, *n* (%): working 42 (45.7), unemployed 50 (54.4). Smoking status, *n* (%): noncurrent 41 (44.6), current 51 (55.4) Health conditions, mean (SD): 1.73 (1.89)	African American	-	Current doctor-prescribed medication use—Yes n (%): 29 (31.5)	mTL, qPCR, Leukocytes	CES-D, HADS	Anxiety, mean (SD): 5.02 (3.86)	Controlling for sociodemographic factors, greater anxiety symptoms were associated with shorter LTL
Starnino et al. [42]	October 2016	Canada	Cohort study		132	45.34 (11.16)	54 (40,9%)	Glasses of alcohol/week, mean (SD): 3.84 (5.38); Smoker *n* (%): 19(14%), BMI: 25.35 (5.00)	NA	Depression	-	RTL, qPCR, Leukocyte	BDI-II, BAI, CRP, MCSD, CMHo	Beck Depression Inventory-II, mean (SD): 7.64 (8.21)	Shorter TL was observed among individuals high in defensiveness and depressive symptoms, as well as in those with less hostility and anxiety. Telomere length is associated with psychological burden though the direction of effect differs depending on the psychological variables under study

List of Abbreviations: bp—base pair; TCA—tricyclic antidepressants; mtDNA—mitochondrial DNA; BMI—Body Mass Index; F—female; M—male; IQR—interquartile range; GAD—Generalized Anxiety Disorder; MDD—major depression disorder; TPO—Thyroid Peroxidase; IL—interleukin; HC—healthy controls; USA—United States of America; SD—Standard Deviation; LTL—Leucocyte Telomere Length.

## Data Availability

Data will be available upon request.

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
