# Peer review of "Telomere Shortening and Psychiatric Disorders: A Systematic Review"

_cells, 2021, doi:10.3390/cells10061423_

Round 1

Reviewer 1 Report

It could be interesting to study the relationship between obesity and inflammation,  and depression and inflammation.

Author Response

We would like to thank Reviewer 1 for the positive comments.

Reviewer 2 Report

In this resubmitted version, authors improved manuscript in few ways as suggested, additional explanations are added in the introduction and especially in the discussion sections.

Some minor changes must be introduced: In the introduction it needs to be clarified that telomeres in humans consist of long double strand DNA repeats TTAGGG (~7-24 kb) and short (~75-200 nucleotides) G-rich 3’-protruding single strand end; I would also suggest mentioning accumulation of senescent cells in tissues, including brain, with aging and consequently increasing senescent-associated secretory phenotype (SASP) of which some components and effects are already described. SASP is not mentioned at all; Further, I believe section 3.4. (Quality assessment) should be presented in the form of table, current textual form is not easy to follow and it may be confusing to some readers.

Author Response

Answers to Reviewer #2’ Comments:

1) In this resubmitted version, authors improved manuscript in few ways as suggested, additional explanations are added in the introduction and especially in the discussion sections.

Some minor changes must be introduced: In the introduction it needs to be clarified that telomeres in humans consist of long double strand DNA repeats TTAGGG (~7-24 kb) and short (~75-200 nucleotides) G-rich 3’-protruding single strand end;

ANSWER: We would like to thank Reviewer #2 for the small and pertinent suggestion. In order to improve our introduction, we have clarified the mentioned details in the first paragraph of the Introduction.

2) I would also suggest mentioning accumulation of senescent cells in tissues, including brain, with aging and consequently increasing senescent-associated secretory phenotype (SASP) of which some components and effects are already described. SASP is not mentioned at all;

ANSWER: We thank you a lot for this suggestion. Indeed, SASP was not mentioned in our previous submission. For this resubmission, we now provide new information regarding SASP and aging in the Discussion section, following the topic 4.3.2 Inflammation, as we believe these topics are correlated.

3) Further, I believe section 3.4. (Quality assessment) should be presented in the form of table, current textual form is not easy to follow and it may be confusing to some readers.

ANSWER: We appreciate this observation. We have already provided three tables that present the details of our Quality Assessment section in our prior submission. However, we have not mentioned the tables in the text form at all, which probably have caused this situation. We apologize for that. In order to clarify that, we inserted indications in the Quality Assessment section that references these three tables.

We would like to thank Reviewer 2 for the suggestions, comments and the efforts to improve our text quality

This manuscript is a resubmission of an earlier submission. The following is a list of the peer review reports and author responses from that submission.

Round 1

Reviewer 1 Report

It is a very interesting article and perfect from the methodological point of view.

To improve the article there are two small suggestions.
First, it is considered interesting to include the link to the registration in Prospero.
Secondly, the relationship between obesity, overweight and mental illness is described in the bibliography as well as the association between telomere length and weight, so it would be interesting to reflect on the possible role of obesity and overweight as a confounding factor in the discussion section.

Reviewer 2 Report

A variety of reviews and meta-analytic studies have previously been published on associations between mental health and telomere length. Indeed, detailed meta-analyses for clinical depression (e.g., https://www.ncbi.nlm.nih.gov/pmc/articles/PMC4760624/), anxiety (e.g., https://www.sciencedirect.com/science/article/abs/pii/S092493381730113X), and PTSD (e.g., https://www.nature.com/articles/s41598-017-04682-w) are readily available. Moreover, systematic reviews of associations between mental disorders and telomere length have been published previously (e.g., https://www.ncbi.nlm.nih.gov/pmc/articles/PMC5778888/), in addition to reviews of stress at various stages of life (e.g., https://pubmed.ncbi.nlm.nih.gov/23639252/; https://www.ncbi.nlm.nih.gov/pmc/articles/PMC3557830/). Furthermore, prior work on the topic includes much more detail on biobehavioral mechanisms explaining the association. Because a large amount of this work already exists, the manuscript does not provide a significant contribution to the literature.

Reviewer 3 Report

Review: Manuscript ID: cells-1111181

Paper “Telomere shortening and psychiatric disorders: a systematic review” submitted by PA Pousa, RM Souza, PHM Melo, BHM Correa, TSC Mendonça, ACSimões-e-Silva and DM Miranda.

The declared aims of this paper are: (1) to perform a systematic review of the existing data; (2) to answer whether it is possible to establish a link between TL and disorders related to psychological distress, including depression, anxiety and PTSD and (3) to unveil the molecular mechanisms behind telomere shortening in mental health disorders.

The authors described in overwhelming detail  a selection of papers for their analysis. In the results section they provide two figures for search queries and a flow diagram of study selection which, I believe is sufficient since their selection procedure is described in great details both in the methodology and results sections. The results are merely a description of selected papers regarding the type of their research. The discussion doesn't bring any novelty since it only shortly describes the selected papers. They emphasize some contradictory results among papers which has been previously described in many other publications, including reviews.

In my opinion, this paper fulfiled only one aim (1) while it fell short of answering aim (2). Most importantly, they provide no explanation relevant to aim (3) „to unveil the molecular mechanisms behind the telomere shortening in mental health disorders” which would be the most interesting contribution to this field of research. Actually, any molecular approach to causality between telomere biology and mental disorders is missing. In this regard, some very important papers are not addressed in the discussion (Epel et al. PNAS Dec. 7, 2004 vol. 101 no. 49; Ornish et al. Lancet Oncol. 2008 Nov;9(11); Ornish et al. Lancet Oncol. 2013 Oct;14(11); Jacobs et al. Psychoneuroendocrinology Vol. 36, Issue 5, 2011) from which it is obvious that physical and mental state, lifestyle, socioeconomic status etc. influence telomere biology. Importantly, by interventions in lifestyle and mental state of an individual telomere shortening dynamics can be manipulated, both in a positive as well as negative way. Therefore, answer to the question mentioned under aim (2) of this paper has already been provided elswhere.

In conclusion, I could not find sufficient novelty in this paper that would reach the quality criteria to be published in Cells.